# PBRM1 loss defines a nonimmunogenic tumor phenotype associated with checkpoint inhibitor resistance in renal carcinoma

Xian-De Liu [1,10 ✉], Wen Kong [1,2,10], Christine B. Peterson [3], Daniel J. McGrail[4], Anh Hoang[1], Xuesong Zhang[1], Truong Lam[1], Patrick G. Pilie[1], Haifeng Zhu[5], Kathryn E. Beckermann[6], Scott M. Haake[6], Sevinj Isgandrova[7], Margarita Martinez-Moczygemba[7], Nidhi Sahni [8], Nizar M. Tannir[1], Shiaw-Yih Lin[4], W. Kimryn Rathmell[9] & Eric Jonasch[1 ✉]

A non-immunogenic tumor microenvironment (TME) is a significant barrier to immune checkpoint blockade (ICB) response. The impact of *Polybromo-1* (*PBRM1*) on TME and response to ICB in renal cell carcinoma (RCC) remains to be resolved. Here we show that *PBRM1/Pbrm1* deficiency reduces the binding of brahma-related gene 1 (BRG1) to the IFNγ receptor 2 (*Ifngr2*) promoter, decreasing STAT1 phosphorylation and the subsequent expression of IFNγ target genes. An analysis of 3 independent patient cohorts and of murine pre-clinical models reveals that PBRM1 loss is associated with a less immunogenic TME and upregulated angiogenesis. *Pbrm1* deficient Renca subcutaneous tumors in mice are more resistance to ICB, and a retrospective analysis of the IMmotion150 RCC study also suggests that *PBRM1* mutation reduces benefit from ICB. Our study sheds light on the influence of *PBRM1* mutations on IFNγ-STAT1 signaling and TME, and can inform additional preclinical and clinical studies in RCC.

[1] Department of Genitourinary Medical Oncology, The University of Texas MD Anderson Cancer Center, Houston, TX 77030, USA. [2] Department of Urology, Renji Hospital, Shanghai Jiao Tong University School of Medicine, Shanghai, China 200127. [3] Department of Biostatistics, The University of Texas MD Anderson Cancer Center, Houston, TX 77030, USA. [4] Department of Systems Biology, The University of Texas MD Anderson Cancer Center, Houston, TX 77030, USA. [5] Department of Neuro-Oncology, The University of Texas MD Anderson Cancer Center, Houston, TX 77030, USA. [6] Division of Hematology and Oncology, Vanderbilt University Medical Center, Nashville, TN 37232, USA. [7] Institute of Biosciences and Technology, Texas A&M Health Science Center, Houston, TX 77030, USA. [8] Department of Epigenetics and Molecular Carcinogenesis, The University of Texas MD Anderson Cancer Center, Smithville, TX 78957, USA. [9] Vanderbilt-Ingram Cancer Center, Division of Hematology and Oncology, Department of Medicine, Vanderbilt University Medical Center, Nashville, TN 37232, USA. [10]These authors contributed equally: Xian-De Liu, Wen Kong ✉email: xliu10@mdanderson.org; ejonasch@mdanderson.org

Cancers of the kidney and renal pelvis afflict over 70,000 individuals and cause approximately 14,000 deaths per year in the USA[1]. Clear cell renal cell carcinoma (ccRCC) is the most common histological subtype of kidney cancer, and has long been recognized to be an immunogenic tumor[2]. The treatment landscape for advanced RCC evolved significantly in the past few years with the approval of immune checkpoint-blocking antibodies, such as nivolumab, pembrolizumab and ipilimumab, which induce profound and durable response in a subset of patients either as monotherapy, as doublets or in combination with antiangiogenic agents[3–5]. The majority of patients, however, still fail to achieve a durable response to immune checkpoint blockade (ICB) due to intrinsic or adaptive resistance. Identifying the drivers of response in the biologically heterogeneous ccRCC patients treated with ICB is in unmet and urgent clinical need[3–5].

Our rapidly evolving understanding of mechanisms underlying ICB response in melanoma has revealed that a non-immunogenic tumor microenvironment (TME) is a significant barrier to immunotherapy benefit[6,7]. Response to ICB requires the presence of antitumor T cells in the TME, while their activity is inhibited by checkpoint pathways[8,9]. This subset of responsive tumors is associated with an immunogenic TME as characterized by the high expression of T cell-inflamed signatures or of IFNγ-related profiles[10,11]. Defects in the IFNγ signaling pathway, specifically mutations in IFNGR1, JAK1, JAK2, and STAT1, induced resistance to ICB in patients with metastatic melanoma[12–14]. In addition, tumor-intrinsic pathways, such as activation of the WNT/β-catenin pathway and loss of PTEN, were associated with resistance to anti-PD-L1/anti-CTLA-4 therapy in metastatic melanoma[11,15]. Therefore, it is conceivable that, in patients with ccRCC, the interaction between fundamental gene mutations in tumor cells and the TME also determines resistance to immunotherapy.

Unlike melanoma, RCC is associated with a low to moderate tumor mutational burden (TMB)[16], and there is no evidence for microsatellite instability in RCC[17]. Most ccRCC cases are associated with genetic deletions and mutations, or epigenetic silencing of the von Hippel-Lindau (VHL) gene, which results in an accumulation of hypoxia-inducible factors that drive dysregulated angiogenesis[18]. CcRCC has a number of secondary mutations, including Polybromo-1 (PBRM1) or BAF180, SET domain-containing 2 (SETD2), and BRCA1 associated protein 1 (BAP1), whose roles in immune modulation remain unclear[16]. PBRM1 or BAF180 is part of the switch/sucrose non-fermenting (SWI/SNF) chromatin remodeling complex, and ~40% of ccRCC have PBRM1 mutations[16,19]. Data thus far on the effect of PBRM1 loss on immune responsiveness are inconsistent. Recently, PBRM1 mutations were reported to be associated with clinical benefit from anti-PD-1 therapy in ccRCC patients who received prior antiangiogenic therapy[20,21]. However, other contemporary studies failed to indicate PBRM1 mutations were a positive predictive biomarker for response to ICB[5,22,23]. It was reported that Pbrm1-deficient murine B16F10 melanomas were more immunogenic and more responsive to immunotherapy[24]. However, the significance of this melanoma model is unclear, since human melanoma tumors rarely harbor PBRM1 mutations[25] and RCC demonstrated distinct immune cell-inflamed signatures that were different than melanoma and most other type of tumors[26]. Thus, RCC-specific mechanistic and clinical data are critically needed to precisely further characterize the influence of PBRM1 loss on response to immunotherapy.

In this study, we found that PBRM1 loss reduced IFNγ-STAT1 signaling in murine and human RCC cell lines, respectively, in a SWI/SNF complex dependent manner. PBRM1 inactivation was associated with a less immunogenic TME and with resistance to immunotherapy in an immunocompetent murine RCC model.

Consistent with these findings, we observed that PBRM1 mutations were associated with decreased immune infiltrates in an analysis of nearly 700 patients with ccRCC, and with poor response to ICB-containing therapy. Taken together, these findings demonstrate that PBRM1 is a key regulator of tumor cell-autonomous immune response in RCC, and loss of PBRM1 function likely contributes to the blunted ICB response experienced by many patients.

## Results

**PBRM1 loss reduced IFNγ-JAK2-STAT1 signaling.** In order to investigate the influence of PBRM1 loss on response to immunotherapy in an immunocompetent RCC model, we generated Pbrm1 knockout Renca murine RCC cell lines using the CRISPR/Cas9 technique. Renca is a broadly used murine RCC cell line, derived from a spontaneously arising tumor in a BALB/c background, and without known Vhl and Pbrm1 mutations. Since constitutive Cas9 expression in Renca cells has previously been shown to induce immune rejection in BALB/c mice[27], we employed a plasmid-based Cas9 knockout system (Santa Cruz®) that resulted in transient Cas9 expression. We identified three clones (#2, #4, and #18) with complete knockout at the protein level and nearly complete at the mRNA level (Fig. 1a, b).

In the absence of IFNγ, IFNGR1 and IFNGR2 are separated and associated with inactive forms of receptor-binding Janus kinase 1 (JAK1) and JAK2, respectively. IFNγ binding induces assembly of an active receptor tetramer, leading to activation of JAKs which phosphorylate signal transducer and activator of transcription 1 (STAT1) on Y701, with subsequent STAT1 homodimerization and nuclear translocation[28]. STAT1 phosphorylation on S727 at the transactivating domain maximizes STAT1 transcriptional activity[29,30]. Since IFNγ target genes are involved in T-cell infiltration, activation and suppression, and thus modulate the TME, we first compared IFNγ-STAT1 activity in Pbrm1 proficient and deficient Renca cells. Following IFNγ stimulation, control cells exhibited a dramatic increase in STAT1 phosphorylation at Y701 and S727, peaking at 2 and 8 h, respectively (Fig. 1c). However, Pbrm1 knockout inhibited IFNγ-induced STAT1 phosphorylation at both time points and phosphorylation sites (Fig. 1c). We observed that the phosphorylation of JAK2, the upstream kinase of STAT1, was also impaired by Pbrm1 knockout, while the total protein level of JAK2 was comparable between cells with or without Pbrm1 expression (Fig. 1c, S1C). In addition, Pbrm1 knockout did not reduce JAK1 phosphorylation (Fig. 1c).

In order to confirm the changes in STAT1 activity, we investigated the expression of IFNγ-induced genes, including transcription factors Stat1 and Irf1[31–33], and chemoattractive factors Cxcl9 and Icam1[34,35]. In treatment naive cells, expression of these genes was low. IFNγ treatment dramatically increased their expression in control knockout cells, while it was largely suppressed in Pbrm1 knockout cells (Fig. 1d, Supplementary Figs. 1A, B). We further confirmed the reduction of STAT1 and IRF1 protein level by western blot (Fig. 1c, S1C), and CXCL9 secretion using an enzyme-linked immunosorbent assay (ELISA) (Fig. 1e). Interestingly, IFNγ-induced Irf1 mRNA expression was relatively fast and transient, peaking at 2-h-treatment and dropping after 8-h-treatment (Supplementary Fig. 1A). However, there was a delay in protein loss since no obvious decrease was observed after 2-h versus 8-h treatment (Fig. 1c). We suspect that IRF1 translation is sustained longer than transcription and/or IRF1 protein is more stable than Irf1 mRNA. These results all confirm that PBRM1 loss inhibited the activity of the IFNγ-JAK2-STAT1 signaling pathway and the expression of downstream target genes.

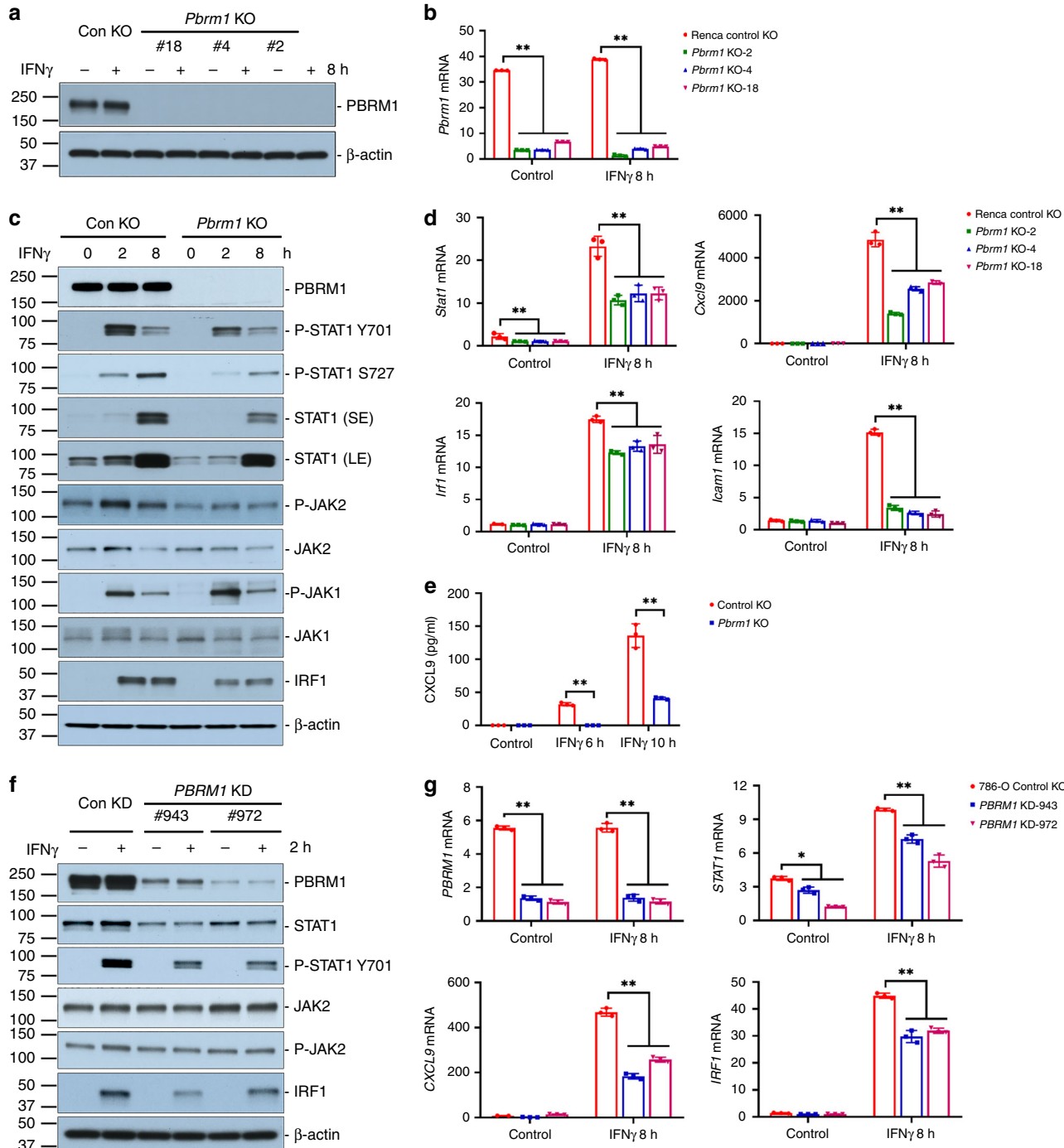

**Fig. 1 *PBRM1/Pbrm1* deficiency reduced IFNγ-STAT1 activity in Renca cells and 786-O cells. a** *Pbrm1* knockout validation in Renca cells at protein levels by western blot, and **b** at mRNA levels by real-time PCR. Renca cell were treated with or without 1 ng/ml IFNγ for 8 h. **c** IFNγ-induced JAK-STAT1 expression and phosphorylation in Renca cells. Control KO or *Pbrm1* KO (clone #18) Renca cells were treated with 1 ng/ml IFNγ for 2 or 8 h. Cell lysates were analyzed by immunoblot using antibodies against PBRM1, STAT1, P-STA1 Y701, P-STAT1 S727, JAK2, P-JAK2 Y1007/1008, JAK1, P-JAK1 Y1034/ 1035, IRF1. β-actin was used an internal control. **d** IFNγ-induced gene expression in Renca cells. Control KO or *Pbrm1* KO (clone #18) Renca cells were treated with 1 ng ml IFNγ for 8 h. mRNA expression of *Stat1*, *Cxcl9*, *Irf1*, and *Icam1* were detected by real-time PCR. *Gapdh* was used as internal control. **e** IFNγ-induced CXCL9 secretion. Renca cells were cultured in serum-free medium and treated with 1 ng/ml IFNγ for 4 or 10 h. The concentration of CXCL9 was analyzed using Quantikine® ELISA kit. **f** IFNγ-induced JAK-STAT1 expression and phosphorylation in 786-O cells. Control knockdown (Con KD) or *PBRM1* knockdown (*PBRM1* KD) 786-O cells were treated with or without 10 ng/ml IFNγ for 2 h. Cell lysates were analyzed by immunoblot using antibodies against PBRM1, STAT1, P-STA1 Y701, JAK2, P-JAK2 Y1007/1008, and IRF1. β-actin was used an internal control. **g** IFNγ-induced gene expression in 786-O cells. 786-O cells were cultured in DMEM with 10% FBS, and treated with 10 ng/ml IFNγ for 8 h. mRNA expression of *STAT1*, *CXCL9*, and *IRF1* were detected by real-time PCR. *GAPDH* was used as internal control. Unpaired *t*-test was performed with GraphPad Prism 7.03. *$P < 0.05$ and **$P < 0.001$, compared with control knockout or knockdown cells. All data are representative of three independent experiments. Data in the bar graphs represent mean ± S.D., $n = 3$. Source data are provided as a Source Data file.

In order to expand the significance of our findings to human RCC, we generated isogenic *PBRM1* knockdown clones in the human 786-O RCC cell line, which is innately *VHL* null and *PBRM1* intact. We observed that mRNA and protein levels in both knockdown clones (#943 and #972) were much lower than that in the control knockdown clone, confirming the efficiency of the *PBRM1* knockdown (Fig. 1f, g). We then assessed the influence of PBRM1 on IFNγ-STAT1 signaling in 786-O cells. Consistent with the results obtained in Renca *Pbrm1* knockout cells, *PBRM1* knockdown also decreased the expression of IFNγ-induced genes in 786-O cells, including *STAT1*, *IRF1*, and *CXCL9* (Fig. 1g). Furthermore, we also confirmed that *PBRM1* knockdown decreased STAT1 total protein and phosphorylation levels, as well as IRF1 protein levels in 786-O cells (Fig. 1f). These results indicate that PBRM1 loss also reduced IFNγ-STAT1 activity and downstream gene expression in human RCC cells.

**PBRM1 loss impaired BRG1 binding to *Ifngr2* promoter**. It has been reported that BRG1, the core ATPase subunit of SWI/SNF complex, is required for STAT1 binding to IFNγ target promoters, such as *CIITA*, *GBP1*, and *IFI27*[36]. We hypothesized that PBRM1 loss reduced the expression of IFNγ target genes by impairing the binding of BRG1 and STAT1 to the promoters. We performed a chromatin immunoprecipitation (ChIP) assay to evaluate the protein–DNA interaction, and immunoprecipitated DNA with appropriate antibodies followed by amplification and quantification by real-time PCR. As we expected, the binding of BRG1 and STAT1 to *Cxcl9* and *Cxcl10* promoters decreased in *Pbrm1* knockout Renca cells (Fig. 2a). Furthermore, BRG1 was also reported to be required for the recruitment of transcription factor SP1 for matrix metalloproteinase 2 (*MMP2*) expression[37]. In the promoter of murine *Ifngr2*, multiple binding motifs for the SP1 transcription factor were predicted[38]. We hypothesized that PBRM1 deficiency might also reduce the binding of BRG1 to the *Ifngr2* promoter. Huang et al. identified a genomic locus with 5-hydroxymethylcytosine (5hmC) enrichment at the proximal promoter region of *Ifngr2*, which was associated with gene transcriptional activity[39]. ChIP assay with antibody against BRG1 revealed that *Pbrm1* knockout reduced the binding of BRG1 to this 5hmC-enriched region in the *Ifngr2* promoter, implying that PBRM1 might help BRG1 binding to *Ifngr2* promoter, assist assembly of chromatin remodeling complex and initiate *Ifngr2* transcription (Fig. 2b). Similarly, the binding of SP1 was also reduced by *Pbrm1* deficiency (Fig. 2b). In order to evaluate the consequence of reduced binding of BRG1 to *Ifngr2* promoter, we confirmed that *Pbrm1* knockout reduced *Ifngr2* expression but did not affect *Ifngr1* expression, in both untreated and IFNγ-treated Renca cells (Fig. 2c). Consistently, *Pbrm1* deficiency reduced IFNGR2 but not IFNGR1 at protein level (Fig. 2d). Furthermore, we confirmed the reduction in IFNGR2 at the cell surface by flow cytometry (Fig. 2e). A negative feedback loop between IFNγ and its receptors has been observed in T helper cells[40,41], here we also observed that IFNγ treatment slightly decreased IFNGR1 and IFNGR2 at both mRNA level and protein level (Fig. 2c, d). Our results collectively indicate that PBRM1 regulates the binding of BRG1 and transcription factors to both IFNγ receptor and target genes. BRG1 inhibition has been reported to reduce BRG1 binding to *CCNB1* and *LTBP2* promoters and decrease repressor element 1-silencing transcription factor (REST)-chromatin interaction[42,43]. It is conceivable that *PBRM1* mutations may functionally alter the SWI/SNF complex and reduce the binding of BRG1 to the *Ifngr2* promoter. This study provides a broader mechanistic understanding of the reduced IFNγ target gene expression in PBRM1-deficient cells.

**Pbrm1 knockout was associated with a less immunogenic TME in murine RCC**. To expand on our observations, we developed a transcriptomic signature to evaluate tumor immunogenicity in RCC based on profiles predicting clinical response to PD-1 blockade in melanoma and urothelial carcinomas[10,44]. This immunomodulatory signature includes genes related to the IFNγ signaling pathway (*IFNG*, *STAT1*, and *IRF1*), antigen presentation (*CIITA* and *HLA-DRA*), T cell recruitment (*CCL5*, *CCR5*, *CXCR6*, *CXCL9*, *CXCL10*, and *ICAM1*), T cell marker and activity (*CD3E*, *CD4*, *CD8A*, *CD28*, *CD80*, *GZMB*, and *PRF1*), and immunosuppressive factors (*CTLA4*, *CD274*, *LAG3*, *PDCD1*, *PDCD1LG2*, and *IDO1*). Employing the Renca-BALB/c immune competent murine model system, we compared Renca control knockout tumors with tumors derived from two *Pbrm1* knockout clones (clone #4 and #18, $n = 5$ each group). Principal component analysis (PCA) revealed that the two *Pbrm1* knockout lines were more similar to each other than to tumors derived from the control knockout, and thus we combined both *Pbrm1* knockout clones as one group (Supplementary Fig. 2). First, we generated a plot of the confidence intervals for the differences in these genes between *Pbrm1* wild-type tumors and those with *Pbrm1*-deficient tumors (Fig. 3a). The x-axis is the difference in group means of *Pbrm1* knockout group minus *Pbrm1* wild-type group, so negative values correspond to genes downregulated in the *Pbrm1* knockout group. We found that the expression of a cytotoxic T cell marker (*Cd8a*), immune checkpoint markers (*Ctla4* and *Pdcd1*), and T-cell chemoattractive factors (*Cxcl10* and *Icam1*) were significantly lower in *Pbrm1* knockout tumors than that in control knockout tumors (Fig. 3a). Most other genes also exhibited concordant directionality, but did not reach statistical significance, possibly due to the limited sample size (Fig. 3a). We further validated the expression of a subset of these genes using real-time PCR. The expression of *Ifng* (not covered in RNAseq), *Cxcl9*, *Cxcl10*, and *Pdcd1* was significantly decreased in *Pbrm1*-deficient tumors (Fig. 3b). As we observed in Renca cell lines, the overall *Pbrm1* mRNA level in *Pbrm1* knockout tumors was much lower than that in control tumors (Fig. 3c). *Pbrm1* knockout significantly reduced the expression of *Ifngr2* but not *Ifngr1* (Fig. 3c). We then applied gene set enrichment analysis (GSEA) to assess whether this set of genes had coordinated differences across tumors with or without *Pbrm1* expression. We found that, as a set, these genes tend to be more highly expressed in *Pbrm1* intact tumors (Fig. 3d).

We then assessed immune markers, including CD3, CD8, CD4, PD-1, and P-STAT1 in murine Renca tumors, using immuno-histochemical (IHC) staining (Fig. 3e). We found that CD3, CD4, CD8 T, and p-STAT1 positive cell levels were significantly higher in control knockout tumors than that in *Pbrm1* knockout tumors (Fig. 3e). Importantly, the expression of immune checkpoint protein PD-1 was also higher in control knockout tumors (Fig. 3e). The Opal multiplex IHC staining results further showed CD8 positive T cells also expressed PD-1, indicating that the activity of infiltrating T cells could be inhibited by immune checkpoint pathways (Fig. 3f). In support of our findings in Renca tumors, we found that *Pbrm1* deficiency in pre-malignant renal cortices of mice[45] reduced the expression of the immunomodulatory profile and the abundance of total T cells and CD8 T cells (Fig. 3g, h). These results collectively indicate that *Pbrm1* deficiency was associated with a less immunogenic microenvironment in different murine models.

**PBRM1 mutations were associated with a less immunogenic TME in human ccRCC**. Next, in order to confirm the above results from Renca tumors, we investigated the influence of PBRM1 on the TME in human ccRCC. There are 442 subjects

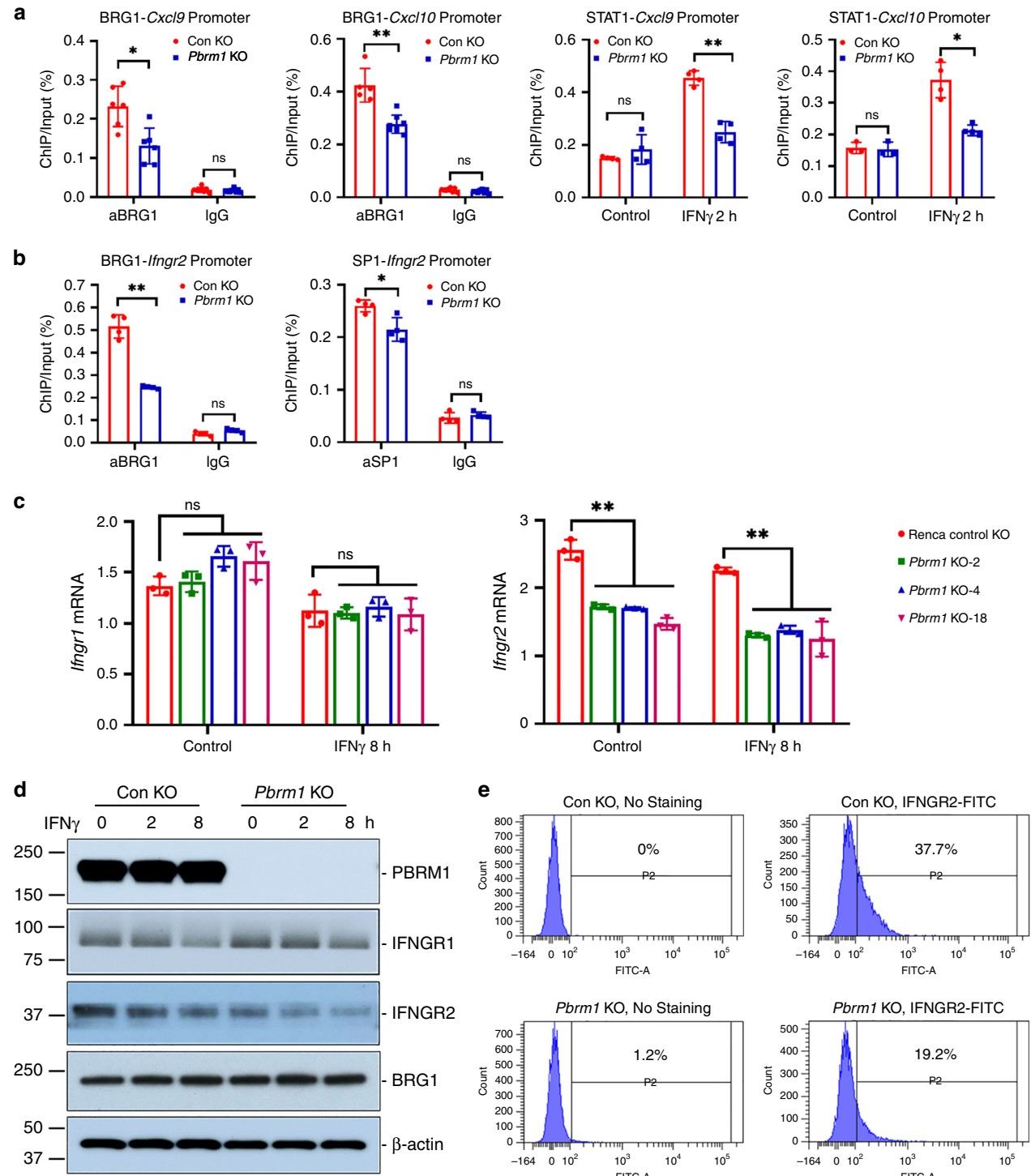

**Fig. 2 *Pbrm1* knockout reduced *Ifngr2* expression in Renca cells. a** BRG1 and STAT1 binding to *Cxcl9* and *Cxcl10* promoter in Renca cells. **b** BRG1 and SP1 binding to *Ifngr2* promoter. Chromatin immunoprecipitation (ChIP) with BRG1, STAT1 or SP1 antibody as indicated in figures was performed using SimpleChIP® Plus Enzymatic Chromatin IP Kit. Isotype IgG was used as negative control. Immunoprecipitated DNA was amplified and quantified by real-time PCR. Protein relative occupancy on promoter was expressed as a percent of the total input chromatin. Con KO or *Pbrm1* KO Renca cells were treated with 1 ng/ml IFNγ for 2 h for IFNγ-induced STAT1 binding to Cxcl9 or Cxcl10 promoter. **c** IFNγ receptor subunits, *Ifngr1* and *Ifngr2*, mRNA expression detected by real-time PCR. *Gapdh* was used as internal control. **d** IFNγ receptor subunits, IFNGR1 and IFNGR2, protein expression was detected by western blot. β-actin was used an internal control. **e** IFNGR2 membrane expression was detected by flow cytometry. Unpaired *t*-test was performed with GraphPad Prism 7.03. *$P < 0.05$ and **$P < 0.001$, compared with control knockout cells. Data in the bar graphs represent mean ± S.D. Source data are provided as a Source Data file.

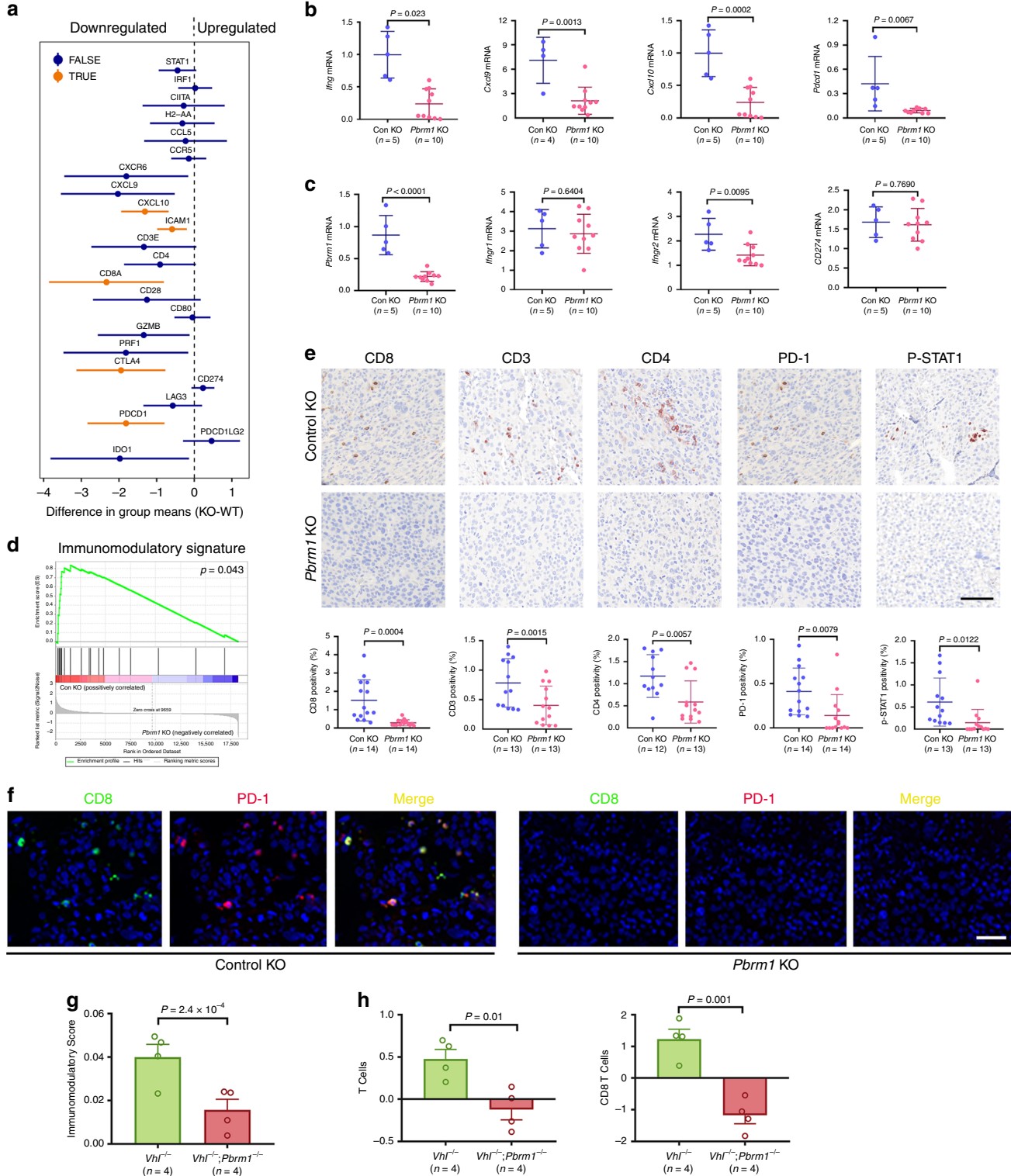

with both mutation and gene expression data (RNA seq) available in TCGA-KIRC dataset. We generated a plot of the confidence intervals for the differences of genes in the same previously mentioned list between *PBRM1* wild-type tumors and *PBRM1* mutated tumors. We found that *PBRM1* mutations were associated with significantly reduced expression of around two thirds of the genes in the immunomodulatory signature (Fig. 4a). Enrichment analysis showed that *PBRM1* mutated tumors exhibited coordinated downregulation of the immunomodulatory gene set relative to the *PBRM1* intact group (Fig. 4b). Similar

reduction was also confirmed in another two human RCC data-sets, IMmotion150 trial cohort[5] and International Cancer Genome Consortium (ICGC) cohort (Fig. 4b). These results indicate the Renca-BALB/c murine model recapitulates immune features seen in human RCC. Furthermore, the expression of multiple predefined immune-related profiles, including IFNγ response, IFNα response, Fcγ receptor signaling pathway, lymphocyte mediated immunity, leukocyte mediated immunity, and adaptive immune response was also reduced in tumors with *PBRM1* mutations across all three RCC patient cohorts (Fig. 4b).

**Fig. 3 *Pbrm1* knockout was associated with less immunogenic TME in murine renal tumor and pre-malignant kidney tissue. a** The confidence interval plots represented the differences in these genes between the two groups (*Pbrm1* KO minus *Pbrm1* WT). The *x*-axis is the difference in group means of *Pbrm1* knockout group minus *Pbrm1* wild-type group, so negative values correspond to genes downregulated in the *Pbrm1* knockout group. The lines depicted 95% confidence intervals, and genes with a significant difference between the groups (adjusted *P*-value <0.05) were marked in orange. The *P*-values were obtained from a two-sample *t*-test on the log2-transformed values, and the resulting *P*-values were adjusted for multiple comparisons using the Benjamini–Hochberg method across the gene set. **b** mRNA expressions of *Ifng*, *Cxcl9*, *Cxcl10*, and *Pdcd1* and **c** the mRNA expression of *Pbrm1*, *Ifngr1*, *Ifngr2*, and *Cd274* were detected by real-time PCR. Each dot represents the mean value of triplicated tumor samples. Unpaired *t*-test was performed with GraphPad Prism 7.03. **d** The coordinated differences of the genes as listed in (**a**) across the two groups assessed by gene set enrichment analysis (GSEA). **e** T cell infiltration and quantification. Murine Renca tumor microarrays, with triplicate formalin-fixed tissue cores for each case, were immunohistochemically stained with antibodies against CD3, CD8, CD4, PD-1, and P-STAT1 Y701. The percentages of positively stained cells were analyzed using inForm software. Unpaired *t*-test was performed with GraphPad Prism 7.03. Scale bar, 100 μm. **f** Multiplex Opal Immunofluorescence staining. The slides were stained with primary antibodies against CD8 and PD-1, corresponding HRP conjugated secondary antibodies, and subsequently TSA dyes to generate Opal signal (CD8, 520 nm; PD-1, 570 nm). PD-1 Opal signals are artificially colored as red. Scale bar, 50 μm. **g** Immunomodulatory gene expression signature score in pre-malignant murine kidneys following loss of *Vhl* alone (*Vhl*$^{-/-}$), or *Vhl* in combination with *Pbrm1* (*Vhl*$^{-/-}$*Pbrm1*$^{-/-}$). Student *t*-test. **h** Gene expression-based inference of total T cell and CD8 T-cell infiltrates in pre-malignant murine kidneys following loss of *Vhl* alone, or *Vhl* in combination with *Pbrm1*. Student *t*-test. Data in the graphs represent mean ± S.D. Source data are provided as a Source Data file.

Next, we investigated the correlation between *PBRM1* mutation and T cell infiltration. According to the IHC intensity in the IMmotion150 dataset, CD8 T cell population was lower in tumors with *PBRM1* mutations than that in *PBRM1* intact tumors (Fig. 4c). To expand this analysis, we used expression-based approaches to infer levels of immune infiltrates, which closely matched CD8 T cells determined by IHC (Fig. 4d). This gene expression-based cell type enrichment analysis revealed that CD8 T cell populations were significantly reduced in *PBRM1* mutated tumors in TCGA, IMmotion150 and ICGC RCC cohorts (Fig. 4e). *PBRM1* mutated tumors also demonstrated reduced PD-L1 expression on immune cells (Fig. 4f). To further investigate the immune landscape of *PBRM1*-deficient tumors, we stained a TMA from 20 RCC patients without prior treatment, including 15 samples with wild-type *PBRM1* and 5 with *PBRM1* mutations, and each sample was triplicated. *PBRM1* mutated tumors were associated with reduced infiltration of CD3 T cells, CD45RO T cells, CD8 T cells, and CD4 T cells (Fig. 4g). The Opal multiplex IHC staining results showed the co-expression of PD-1 in a subset of CD8 positive T cells (Fig. 4h). In support of this finding, we observed that the gene expression of *CD8A* positively correlated with the *PDCD-1* expression in TCGA, IMmotion150 and ICGC RCC cohorts (Supplementary Fig. 3A). We further validated our finding using a TMA from sunitinib-treated primary RCCs, including 12 samples with wild-type *PBRM1* and 10 with *PBRM1* mutations. We confirmed that *PBRM1* intact tumors demonstrated significantly higher CD3 T cells, CD45RO T cells, and CD8 T cells as well (Supplementary Fig. 3B). Furthermore, the percentage of phospho-STAT1 positive cells and PD-1 positive cells was also more abundant in *PBRM1* intact tumors (Supplementary Fig. 3B). PD-L1 positivity was numerically lower in tumors harboring *PBRM1* mutations, although this did not reach statistical significance (Supplementary Fig. 3B). These results indicated that PBRM1 intact tumors were associated a more immunologically active TME than *PBRM1* mutated tumors, which was mirrored by the Renca RCC model system.

Though we found a consistently decreased immunogenic TME in ccRCC patients with PBRM1 loss across 5 patient cohorts, a previous Cas9-based screen in murine melanoma cells indicated that loss of PBRM1 may increase tumor immunogenicity[24]. To attempt to contextualize this previous work with our current findings, we analyzed the effects of PBRM1 loss in four additional TCGA datasets with sufficient sample size (microsatellite stable endometrial cancer, microsatellite instable endometrial cancer, gastric cancer, and bladder cancer). In contrast to ccRCC, these four tumor populations all demonstrated increased immune infiltration in tumors with loss of PBRM1 (Supplementary Fig. 4).

This result suggests that PBRM1 loss has effects on the immune microenvironment that vary based on tumor lineage.

**PBRM1 loss was associated with a more angiogenic TME in human and murine RCC.** In addition to chemoattractive chemokines, aberrant vasculature in tumors also influences T-cell infiltration[46]. We investigated the influence of PBRM1 on angiogenesis in our model system. We generated an angiogenesis signature, including genes encoding VEGFs and PDGFs (*VEGFA*, *VEGFB*, *VEGFC*, *PDGFA*, *PDGFB*, and *PDGFC*) and their receptors (*FLT1*, *FLT4*, *KDR*, and *PDGFRB*), *HIF1A* and HIF target genes (*TGFA*, *LDHA*, *LDHB*, *LDHC*, *CDKN1A*, *CDKH1B*, *CDKN2A*, *SLC2A1*, and *EPO*). We observed that genes from the immunomodulatory gene set demonstrated an overall positive correlation amongst themselves, but an overall negative correlation with the expression of the angiogenesis gene signature (Fig. 5a). CD31 IHC staining in the IMmotion150 cohort revealed that the endothelial cell population was higher in *PBRM1* mutated tumors than that in *PBRM1* intact tumors, which could be accurately recapitulated using the angiogenesis gene expression signature in order to analyze additional patient cohorts (Fig. 5b, c). In all three RCC patient cohorts (TCGA, IMmotion150 and ICGC), *PBRM1* mutations were associated with a higher angiogenesis score (Fig. 5d). A similar increase in the angiogenesis score was also observed in *Pbrm1*-deficient pre-malignant mouse renal cortices (Fig. 5e). In addition, we confirmed that *Pbrm1* KO tumors demonstrated a higher density of CD31 positive cells (Fig. 5f). These findings indicate that PBRM1 inactivation led to upregulated angiogenesis in RCC, which is consistent with previous reports[20,45,47,48], and downregulated immunomodulation. We assumed that either PBRM1 has differential effects on transcription of these two gene groups, or aberrant vasculature together with impaired IFNγ signaling contributes to the generation of a less immunogenic TME. Interestingly, PBRM1 loss was associated with a decreased angiogenesis score in gastric cancer, and a similar trend was observed in endometrial and bladder cancers (Supplementary Fig. 4). These findings further confirm that PBRM1 influences the RCC TME and angiogenesis differently from other tumor lineages.

**PBRM1 loss was associated with resistance to immune checkpoint blockade.** Tumor response to ICB is generally enhanced by the presence of an immunogenic TME, as characterized by pre-existing T cells and an upregulation of immune checkpoint pathways[8,9]. Since *Pbrm1* knockout reduced IFNγ-STAT1 activity, T-cell infiltration and PD-1 expression in our murine system,

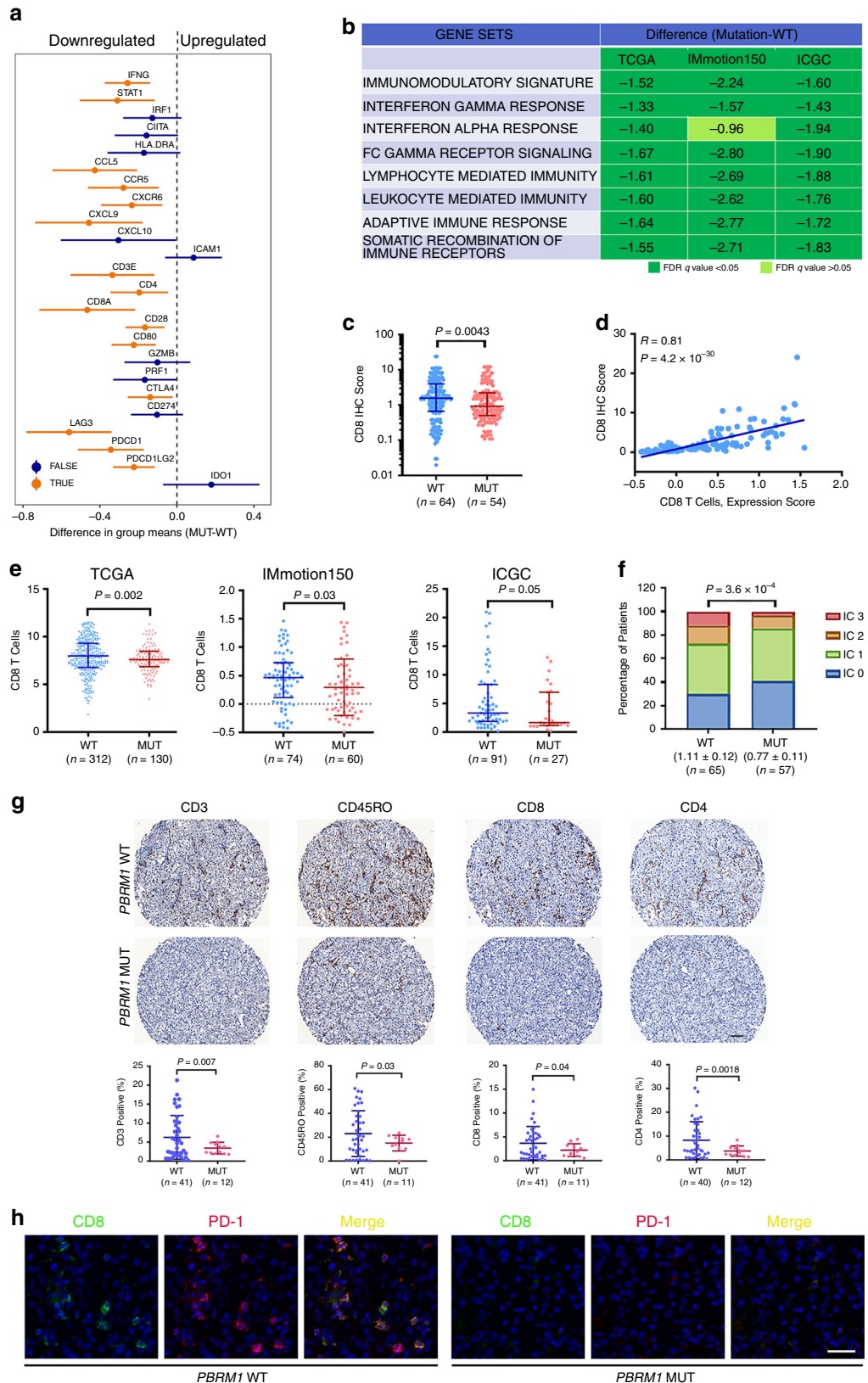

we evaluated the influence of PBRM1 loss on response to ICB. Following previously described treatment algorithms[49], anti-PD-1 (Murine IgG$_{2a}$ clone RMP1-14, BioXcell®) monoclonal antibody was administered on day 3, 6, and 9 post tumor inoculation

(Fig. 6a, Schema). In untreated cohorts, Renca control knockout tumors grew quickly and all mice were sacrificed within 20 days after tumor inoculation due to tumor growth; however, *Pbrm1* knockout tumors grew significantly slower and survived longer

**Fig. 4 PBRM1 loss was associated with less immunogenic TME in human RCC tumors. a** The confidence interval plots represented the differences in immunomodulatory genes between the two groups (mutant *PBRM1* minus wild-type *PBRM1*) in TCGA KIRC dataset. MUT, mutant *PBRM1*; WT, wild-type *PBRM1*. The lines depicted 95% confidence intervals, and genes with a significant difference between the groups (adjusted *P*-value <0.05) were marked in orange. The *P*-values were obtained from a two-sample *t*-test on the log2-transformed values, and the resulting *P*-values were adjusted for multiple comparisons using the Benjamini–Hochberg method across the gene set. **b** Comparison of gene expression in tumors with mutant *PBRM1* (MUT) versus wild-type *PBRM1* (WT) by GSEA using the immunomodulatory genes as listed in (**a**) and other predefined immune-related gene sets in the TCGA, IMmotion150 and ICGC patient cohorts. **c** Comparison of CD8 infiltrates as assessed by immunohistochemistry for CD8 in the IMmotion150 patient cohort. Rank-sum test. **d** CD8 immunostaining levels positively correlated with gene expression-based inference. Spearman correlation and associated *P*-value inset. **e** Gene expression-based inference of CD8 T cell infiltrates in patients stratified by *PBRM1* mutation status. Rank-sum test. **f** Comparison of immune cell PD-L1 expression in patients from the IMmotion150 cohort stratified by *PBRM1* mutation status. Cochran–Armitage test for trend. **g** Treatment naive RCC tumor microarray from 20 untreated RCC patients, including 15 samples with wild-type *PBRM1* and 5 with *PBRM1* mutations, were immunohistochemically stained with antibodies against CD3, CD45RO, CD8, and CD4. Each tumor was triplicated, and the *n* values indicate the number of the intact cores. The percentages of positively stained cells were analyzed using inForm software. Non-parametric Mann–Whitney test was performed with GraphPad Prism 7.03. Scale bar, 100 μm. **h** Multiplex Opal immunofluorescence staining. The slides were stained with primary antibodies against CD8 and PD-1, corresponding HRP conjugated secondary antibodies, and subsequently TSA dyes to generate Opal signal (CD8, 520 nm; PD-1, 620 nm). Scale bar, 50 μm. Data in the graphs represent mean ± S.D. Source data are provided as a Source Data file.

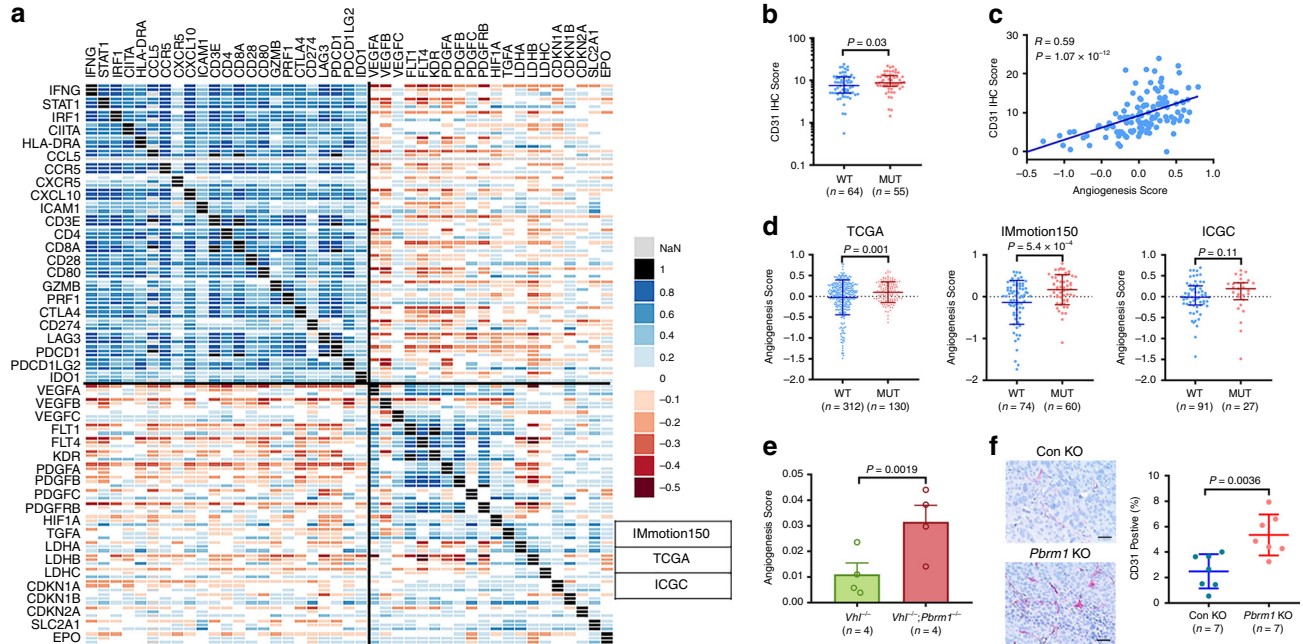

**Fig. 5 Loss of PBRM1 function was associated with increased angiogenesis. a** Correlogram between the expression of immunomodulatory and angiogenic genes IMmotion 150 cohort (top line of each square), TCGA cohort (middle line of each square) and ICGC cohort (bottom line of each square). **b** Patients with mutant *PBRM1* demonstrated increased CD31 immunostaining levels in the IMmotion150 cohort. Rank-sum test. **c** CD31 immunostaining levels positively correlated with angiogenesis expression score. Spearman correlation and associated *P*-value inset (*N* = 119). **d** *PBRM1* mutated tumors were associated with increased angiogenesis score in 3 indicated patient cohorts. Rank-sum test. **e** *Pbrm1* knockout was associated with increased angiogenesis score in pre-malignant murine kidneys. Student *t*-test. **f** *Pbrm1* knockout Renca tumors demonstrated increased CD31 immunostaining levels. Student *t*-test. Data in the bar graphs represent mean ± S.D. Source data are provided as a Source Data file.

(Fig. 6a, Supplementary Fig. 5A). When comparing the changes with or without treatment, anti-PD-1 treatment provided more significant survival benefit and tumor growth control in mice bearing control knockout tumors than in those with *Pbrm1* knockout tumors (Fig. 6a, Supplementary Fig. 5A). Our Renca subcutaneous tumor model confirmed that PBRM1 inactivation slowed tumor progression, while *Pbrm1* knockout tumors were more resistant to early treatment with PD-1 antibody.

We then assessed response to a delayed anti-PD-1 regimen, in which anti-PD-1 treatment was initiated when tumors reached 100–200 mm³. PD-1 blockade significantly reduced tumor growth of control knockout tumors and prolonged survival of mice bearing control knockout tumors, but did not exert significant changes in mice harboring *Pbrm1* knockout tumors (Fig. 6b, Supplementary Fig. 5B). These results further indicate that *Pbrm1*

knockout tumors were also resistant to delayed anti-PD-1 treatment.

To confirm the influence of PBRM1 loss on response to ICB treatment in the clinical setting, we analyzed the IMmotion150 dataset and observed that patients with *PBRM1* mutations demonstrated a significantly lower response rate to atezolizumab (anti-PD-L1) monotherapy or combination therapy with bevacizumab (anti-VEGF) (Fig. 6c, d). We further assessed outcome in ICB-treated RCC patients from the MSK-IMPACT cohort[50]. We found that patients with *PBRM1* mutations demonstrated a shorter overall survival than those with intact *PBRM1* (Fig. 6e). In patients from TCGA, predominately collected from non-ICB-treated patients with ccRCC, *PBRM1* mutations conferred a non-significant trend towards improved survival (Fig. 6f). On further analysis, *PBRM1* mutations were associated with significantly

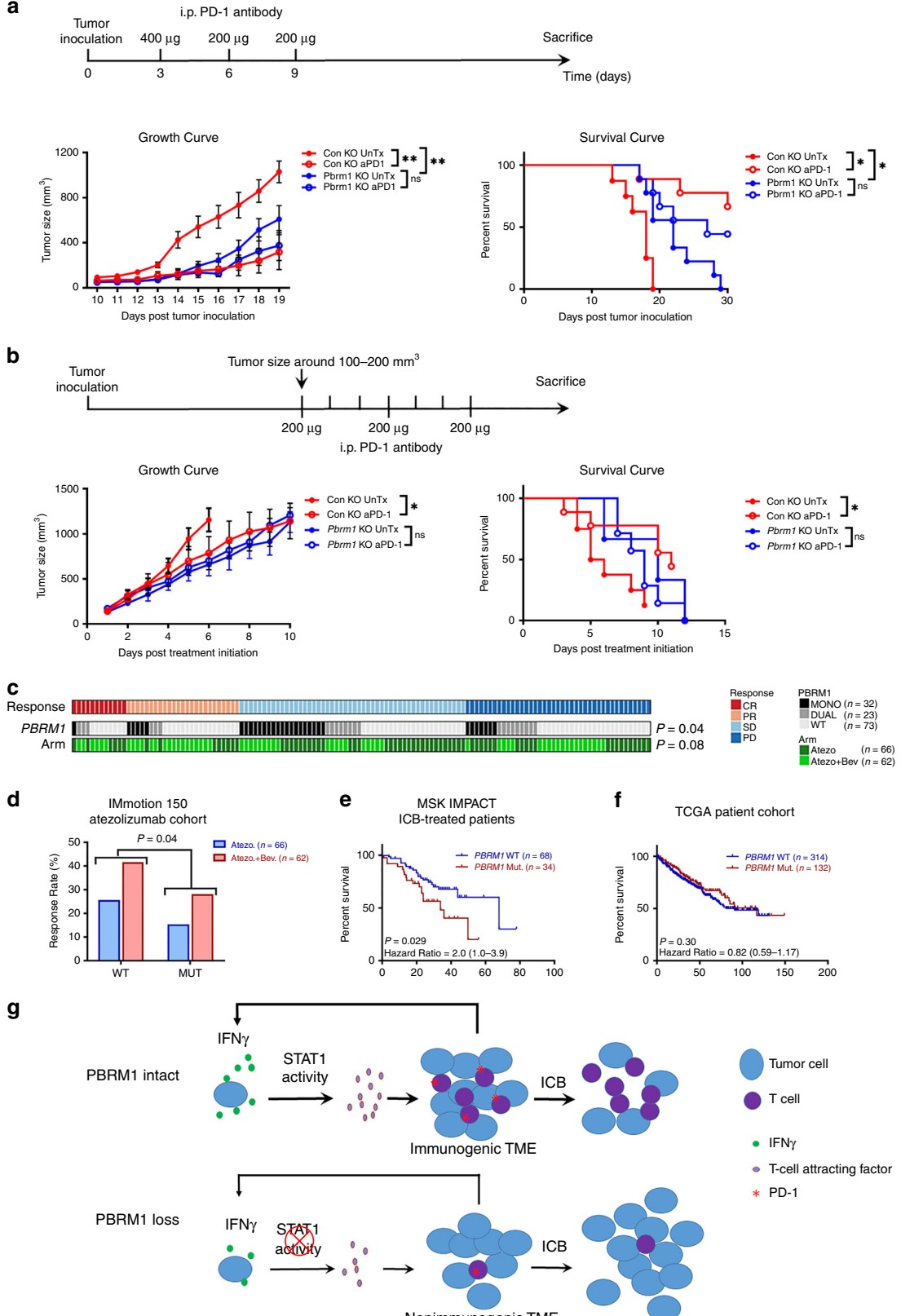

prolonged overall survival in TCGA if the few patients also harboring *BAP1* mutations were excluded from the cohort[51]. These retrospective analysis results collectively suggest that the inferior outcomes of ICB-treated patients with *PBRM1* mutated tumors was not due to a generally poorer prognosis.

Taken together, the data from our isogenic *Pbrm1* knockout murine tumors and pre-malignant mouse kidneys, along with patient cohorts analyzed by IHC (IMmotion150, 2 internal cohorts), gene expression (TCGA, ICGC, IMmotion150), and clinical outcomes following ICB (IMmotion150, MSKCC

**Fig. 6 PBRM1 loss induces resistance to ICB. a** Early treatment with anti-PD-1 blockade. PD-1 antibodies were administrated at day 3, day 6, and day 9 after tumor inoculation. First dose was 400 μg/mouse, and the following two doses were 200 μg/mouse. **b** Delayed treatment with anti-PD-1 blockade. Anti-PD-1 antibody (200 μg/mouse) was administrated every third day once the tumors reached 100-200 mm$^3$. Treatment schemas, In vivo tumor growth rates and survival rates of mice are shown. Two-way ANOVA and Log-rank (Mantel-Cox) analyses were performed with GraphPad Prism 7.03. Data in the graphs are means ± SEM. UnTX, untreated control. ns, $P > 0.05$, $*P < 0.05$, and $**P < 0.001$. **c** Patient response in the IMmotion150 cohort following treatment with either atezolizumab (Atezo) or atezolizumab in combination with bevacizumab (Atezo + Bev). Mono, *PBRM1* mutation only, Dual, *PBRM1* mutation in combination with a *BAP1* or *SETD2* mutation; WT, wild type; CR, complete response; PR, partial response; SD, stable disease; PD, progressive disease. Cochran–Mantel–Haenszel test. **d** Comparison of patient response rate, defined as either a complete or partial response, in the IMmotion150 cohort. Cochran–Mantel–Haenszel test. **e** Overall survival of RCC patients treated with ICB from the MSKCC IMPACT cohort stratified by *PBRM1* status. Log-rank test. **f** Overall survival of ccRCC patients from TCGA stratified by *PBRM1* mutation status. Log-rank test. **g** Model for PBRM1 mediated IFNγ-STAT1 signaling and tumor immune microenvironment modulating. PBRM1 ensures IFNγ-induced STAT1 activity and autonomous expression of downstream genes involved in T-cell recruitment (e.g., *CXCL9*). Infiltrating, activated T cells in turn produce more IFNγ which stimulates tumor cells to secrete immunostimulatory chemokines and cytokines, and in parallel upregulate checkpoint pathways on T cells and tumors cells. Thus, the immunogenic TME of *PBRM1* proficient tumors is primed to respond to ICB. On the other hand, PBRM1 loss reduces IFNγ-STAT1 signaling and downstream T cell attracting factors, which prevents T cell infiltration and IFNγ secretion. Such a non-immunogenic TME of *PBRM1* mutated tumors blunts ICB response. Source data are provided as a Source Data file.

IMPACT) indicate that PBRM1 loss is associated with an immunologically colder TME leading to ICB resistance.

## Discussion

We employed an isogenic murine RCC model to investigate the impact of PBRM1 loss on tumor-autonomous IFNγ signaling and on the TME, measured the effect of PBRM1 loss on response to ICB, and validated our findings in human ccRCC datasets. There are several novel findings in the current study. First, PBRM1 inactivation reduced IFNγ-STAT1 activity and impaired target gene expression. Second, PBRM1 inactivation was associated with a less immunogenic TME in murine tumors, which was confirmed across multiple human ccRCC datasets. Third, PBRM1 inactivation induced resistance to ICB in a Renca-BALB/c immune competent RCC model which could be recapitulated in a ccRCC patient cohort. Taken together, these findings indicate that PBRM1 plays an important role in the IFNγ-STAT1 signaling pathway in RCC, which has divergent effects on the TME. In *PBRM1* wild-type tumors, IFNγ induces the tumor cell-autonomous expression of STAT1, IRF1 and of chemoattractive chemokines, which enhances T-cell infiltration and activation. These activated T cells in turn secrete more IFNγ to stimulate tumor cells, and upregulate checkpoint pathways on T cells and tumors cells as well. Thus, the immunogenic TME is primed to respond to ICB (Fig. 6g). PBRM1 loss reduces IFNγ-induced expression of chemoattractive signals, T-cell infiltration and also IFNγ secretion. Such an immunologically "cold" TME is less responsive to ICB due to the absence of effector T cells (Fig. 6g).

Here we found that *Pbrm1* knockout Renca tumors were associated with a less immunogenic TME, as characterized by the reduced T-cell infiltrations and decreased immunomodulatory gene expression (Fig. 3a–f). Similar reductions were also observed in *Pbrm1* knockout pre-malignant murine kidneys (Fig. 3g, h). Suppression of immunostimulatory gene expression with loss of PBRM1 was confirmed by analysis of nearly 700 human RCC tumors across three independent cohorts, which corresponded with decreased immune infiltrates as analyzed by either gene expression profiling or immunohistochemistry, including two additional patient cohorts (Fig. 4g, Supplementary Fig. 3B). Using an orthogonal expression-based approach, *PBRM1* mutated ccRCC tumors were enriched in a non-inflamed cluster, while *BAP1* mutated tumors were enriched in an immune cell-inflamed cluster[52]. Miao et al. reported the *PBRM1* mutations correlated with increased IL6-JAK-STAT3 signaling[20], which would predict a less T cell-inflamed TME since STAT3 activation inhibits STAT1-dependent gene expression[53,54]. Taken together, PBRM1

loss defines a less immunogenic RCC tumor phenotype. Conversely, a Cas9-screen based approach in melanoma cells previously implicated PBRM1 loss enhanced cytokine expression, a more immunogenic TME and greater response to ICB[24], and similarly we found that other tumor lineages did exhibit immunostimulatory effects following PBRM1 loss (Supplementary Fig. 4). These findings indicate that ccRCC may be unique compared with other cancers with respect to TME modulation and immunotherapy response, and that immunosuppression from PBRM1 loss is unique to ccRCC. PBRM1 has been shown to have pleiotropic effects at a molecular level, including the recognition of acetylated histones and p53, while different bromodomains can either enhance or attenuate nucleosome interaction[55,56]. Tissue specific cells are programmed to express a set of genes unique to that cell type, and each tumor type has additional canonical mutations specific to that tumor. These varied complex interactions can impact positive and negative feedback loops in a way that can produce paradoxical effects. These distinct features of the ccRCC tumor immune microenvironment imply that studies from other tumor lineages cannot be simply translated to RCC, and it is necessary to employ RCC-specific models.

Nevertheless, the influence of PBRM1 on immunotherapy response in RCC patients remains controversial. PBRM1 loss was previously linked to better ICB response in ccRCC[20,21]. Several additional studies failed to find an association between functional PBRM1 loss and clinical benefit from immunotherapy[5,22,23]. In our study, further analysis of patients from the IMmotion150 trial treated with anti-PD-L1 or plus bevacizumab revealed a decreased response rate in tumors with *PBRM1* mutations (Fig. 6c, d). The MKSCC IMPACT study provides additional supportive data (Fig. 6e). All patients in this study had received an ICB at some point in their treatment course, and the presence of a *PBRM1* mutation was associated with worse survival. Our assessment of the ccRCC TCGA KIRC cohort (Fig. 6f) shows that the presence of a *PBRM1* mutation is a neutral or positive prognostic feature, an observation that has been confirmed in other studies[51]. There are several potential possible explanations for these contradictory findings. First, in a nonrandomized study, we cannot assess the interaction between the treatment and biomarker[57], and the favorable prognosis of patients with *PBRM1* mutations could be inappropriately interpreted as a predictive effect following ICB treatment. As noted above, *PBRM1* mutations were reported to be associated with a better prognosis in ccRCC patients[51] and conditional knockout of *Pbrm1* in renal tubule epithelial cells was associated with lower grade tumors with clear cell pathological characteristics in a murine model[58].

Similarly, we observed that *Pbrm1* knockout delayed tumor growth and prolonged survival in our model system. A similar interaction between prognostic and predictive factors is seen in patients with metastatic RCC treated with antiangiogenic therapy, with better prognosis patients typically showing better survival after systemic therapy[59]. Second, the benefit from prior anti-angiogenic therapy could have a continued effect during second-line ICB treatment, or could influence ICB response. *PBRM1* mutated tumors have been reported to be more responsive to antiangiogenic therapy[5]. Although tumor revascularization in model systems occurs fairly rapidly after antiangiogenic therapy withdrawal in model systems[60], clinicopathological data from tumors treated with antiangiogenic therapy show an increase in T-cell infiltration, which is predicted to potentiate response to subsequent ICB treatment[2,61]. In a clinical validation study, Braun et al. reported *PBRM1* mutated tumors were more responsive to ICB in patients who previously received anti-angiogenic therapy[21]. We cannot exclude the potential direct and indirect favorable influence of prior antiangiogenic therapy on ICB response in *PBRM1* mutated tumors. Third, *PBRM1* mutations were associated with resistance to atezolizumab, an anti-PD-L1 antibody in the IMmotion 150 study, but with response to nivolumab, an anti-PD-1 antibody in the study by Brown et al.[5,20,21]. PD-1 blockade interrupts interactions with PD-L1 as well as PD-L2. PD-L1 can also promote cancer cell survival via PD-1 independent pathways[62]. We cannot rule out the possibility that tumors harboring a *PBRM1* mutation may respond to anti-PD-1 and anti-PD-L1 antibodies differently. Forth, the heterogeneity of patients in different cohorts might lead to discordant results. Although these studies focused on the impact of *PBRM1* mutations, ccRCC patients also harbor other secondary mutations, such as *BAP1* and *SETD2*, which might affect ICB response in different ways. It is difficult to precisely isolate the influence of *PBRM1* loss on the tumor microenvironment in the clinical arena due to these variables. For this reason, immune competent, isogenic animal tumor models can help provide some clarity. Our isogenic Renca murine model demonstrates that PBRM1 deficiency blunted the response to ICB treatment, which supports the findings in the IMmotion150 cohort. Importantly, we found that PBRM1 deficiency was associated with a less immunogenic TME in both Renca tumors and human RCC tumors, which is consistent with the widely accepted concept that non-immunogenic tumors are more resistant to ICB therapy[7,63]. *Pbrm1* deficiency in Renca tumors recapitulated the immune features in human RCC.

In summary, this study employed an isogenic murine system with subsequent validation in multiple patient cohorts to demonstrate that PBRM1 loss decreases IFNγ dependent signaling and tumor immunogenicity, and suggest that PBRM1 mutation associates with ICB resistance. The immune competent murine model presented here may assist in the development of therapeutic strategies that can improve T-cell infiltration and immunotherapy response in *Pbrm1* knockout tumors, and guide the management of patients with *PBRM1* mutated tumors.

## Methods

**Antibodies and reagents**. PBRM1 antibody (A301-591A) was from Bethyl Laboratories. Phospho-STAT1 antibody (clone ST1P-11A5; Tyr701; 33-3400), human CD3 antibody (clone F7.2.38; MA5-12577), human CD45RO antibody (clone UCHL1; MA5-11532), human CD4 antibody (clone 4B12; MS1528S0), and human CD8 antibody (clone C8/144B; MS457S0) were from ThermoFisher Scientific. Mouse CD3 antibody (D4V8L; 99940), mouse CD8 antibody(D4W22; 98941), mouse CD4 antibody (D7D2Z; 25229), human PD-L1 antibody (E1L3N; 13684), mouse PD-L1 antibody (D5V3B; 64988), mouse PD-1 antibody (D7D5W; 84651), PBRM1 antibody (D3F7O, 91894), JAK1 antibody (6G4, 3344), JAK2 antibody (D2E12, 3230), Phospho-JAK1 antibody (D7N4Z, Tyr1034/1035, 74129), Phospho-JAK2 antibody (C80C3, Tyr1007/1008, 3776), STAT1 antibody (D1K9Y, 14994), Phospho-STAT1 antibody (58D6, Tyr701, 9167), Phospho-STAT1 antibody (D3B7, Ser727, 8826), IRF1 antibody (D5E4, 8478), and BRG1 antibody

(E9O6E; 52251) were from Cell Signaling Technology. IFNGR1 antibody (112802) was from BioLegend. IFNGR2 antibody (Cat No. GTX 64548) was from GeneTex. β-actin antibody (A1978) was from Sigma. Human IFNγ (285-IF-100) and mouse IFNγ (8234-MB-010) were from R&D Systems.

**Cell culture and establishment of *Pbrm1* knockout or *PBRM1* knockdown cell lines**. Renca cells and 786-O cells were maintained in DMEM containing 10% fetal bovine serum at 37 °C in a humidified incubator with 5% $CO_2$ in a humidified incubator. STR DNA fingerprinting of 786-O cells was performed by the CCSG-funded Characterized Cell Line Core, NCI # CA016672. Renca cells were transfected with control knockout CRISPR/Cas9 plasmid (Santa Cruz Biotechnology, sc418922) or murine *Pbrm1* CRISPR/Cas9 knockout plasmids (sc-426270) plus homology-directed repair (HDR) plasmid (sc-426270-HDR). Cell transfection were mediated with Lipofectamine 2000 (Invitrogen). HDR Plasmid expresses puromycin resistance gene to enable selection of stable knockout (KO) cells and RFP expression for single-cell sorting. Transfected Renca cells were selected in medium containing 2 µg/ml puromycin, and then single knockout clones were sorted using BD FACS FUSION flow cytometer. 786-O stable cell lines expressing control shRNA or *PBRM1* shRNA (Dharmacon, V3LHS_318943 and V2LHS_174972) were infected with lentiviral particles and selected in medium containing 2 µg/ml puromycin.

**Cell lysis and immunoblot analysis**. For immunoblot analysis using total cell lysates, cells were lysed on ice for 30 min in RIPA buffer (50 mM Tris-Cl, pH 7.4, 150 mM NaCl, 2 mM EDTA, 1% Nonidet P-40, 0.1% SDS). Protease inhibitor mixture (BD Biosciences Pharmingen) and Benzonase Nuclease (Vovagen) were added to cell lysis buffer. Proteins were detected with specific primary antibodies and subsequently secondary antibodies.

**RNA isolation and real-time PCR**. Total RNAs were isolated and purified using the RNeasy Mini Kit (Qiagen, 74106) and converted to cDNA using iScript[TM] Reverse Transcription Supermix (Bio-Rad, 1708841). mRNA expression was measured using a real-time PCR detection system (Applied Biosystems ViiA 7) in 96-well optical plates using SsoAdvanced Universal SYBR Green supermix (Bio-Rad, 1725275). *GAPDH/Gapdh* was used as a control. Primer sequences were attached in Table S1.

**Chromatin immunoprecipitation**. Chromatin immunoprecipitation (ChIP) were performed using SimpleChIP® Plus Enzymatic Chromatin IP Kit (Cell Signaling Technology, 9005) according to enclosed chromatin immunoprecipitation protocol. Briefly, cells were incubated with 1% formaldehyde for 10 min at room temperature to cross-link proteins to DNA, nuclei were digested with micrococcal nuclease and sonication. Cross-linked and digested chromatin was immunoprecipitated using indicated antibodies. Immunoprecipitated chromatin was incubated with 5 M NaCl and Proteinase K at 65 °C to reverse cross-links and followed by DNA purification. DNA was quantified by real-time PCR using respective primers. Primer sequences were attached in Table S1.

**ELISA**. Cells ($1 \times 10^6$) were plated in 6-well plates with complete growth medium and cultured overnight. On the following day, cells were washed with serum-free medium three times and treated with 1 ng/ml IFNγ for indicated duration. Mouse CXCL9/MIG concentration in the supernatants was analyzed using Quantikine® ELISA kit (R&D Systems, MCX900) according to the manufacturer's protocol. Optical density was determined using microplate reader set to 450 nm. The reading at 540 nm was used for wavelength correction.

**Flow cytometry**. Cells were collected and stained with IFNGR2-FITC antibody on ice for 20 min in the dark, and then fixed with 400 µL 4% formaldehyde on ice for 15 min. BD Biosciences LSRII flow cytometer was used for data collection and FACSDiva 6.2 instrument software was used for data analysis. Live and healthy cells were gated by FSC and SSC to remove dead cells with high SSC and debris in the lower left quadrant of dot plot (Supplementary Fig. 6). Unstained negative control cells were used to establish the boundary of the negative signal in the FITC channel (Supplementary Fig. 6). For the IFNGR2-FITC antibody labeled samples, any fluorescent events to the right of the negative gate were considered positive and measured on a percentage basis.

**Mouse experiment**. The animal protocols were approved by Institutional Animal Care and Use Committee (IACUC) of The Health Science Center, Texas A&M University. Four to 6-week-old female BALB/c mice were purchased from TACONIC. $5 \times 10^5$ Renca cells were suspended in 100 µl Matrigel Matrix (Corning, 354234) diluted with PBS at 1:1, and subcutaneously injected into the backs of mice. Mice were left untreated or treated with PD-1 antibody (100 µg/mouse/ 3 days) via intraperitoneal injection (I.P.) for three times every third day. After the tumors were palpable (i.e., tumor volume reached around 100 mm³), tumors were measured every day with caliper, and volume calculations were obtained using the formula $V = (W^2 \times L)/2$. Mice were sacrificed once the tumors reached 1000 mm³, ulceration occurred, or animals showed signs of distress. Tumors were fixed in RNA stabilization reagent for RNA extraction or in 10% buffered formalin phosphate for IHC or Opal staining.

**Tissue microarray, immunohistochemistry, and multiplex opal immuno-fluorescence.** Human subject protocol (2007-0511) was approved by Institutional Research Board at M.D. Anderson Cancer Center. Tissue microarrays (TMA) were generated using a Beecher instrument with 0.6 mm cores taken from the donor block and placed into the recipient block in triplicate for each case. Tissue microarrays with triplicate cores for each case were generated from primary RCC. Murine tumor TMA were generated with 5 mm cores. TMAs were immunohistochemically stained and analyzed using inForm software (Caliper Life Sciences). The slides were stained using Opal 4-color IHC Kit (NEL794001KT) from Perkin Elmer. Microwave treatment (MWT) was applied to perform antigen retrieval, quench endogenous peroxidases, and remove antibodies from earlier staining procedures. Perkin Elmer AR6 Antigen retrieval buffer (pH 6) was used for CD8 and PD-1 staining while Perkin Elmer AR9 Antigen Retrieval buffer (pH 9) was used for P-STAT1 staining. The slides were stained with primary antibodies against CD8 and PD-1, corresponding HRP conjugated secondary antibodies, and subsequently TSA dyes to generate Opal signal (CD8, Opal 520; PD-1, Opal 570, or Opal 620). The slides were scanned with the Vectra image scanning system (Caliper Life Sciences), and signals were unmixed and reconstructed into a composite image with Vectra inForm software 2.4.6.

**Gene expression analysis.** In TCGA KIRC, there are $n = 442$ subjects in this dataset with both mutation and gene expression data (RNA seq) available. The expression data is available for 20505 genes. The data was imported into R using the TCGA2STAT package[64]. In murine RNAseq dataset, the comparison is of the Renca control knockout lines ($n = 5$) to $Pbrm1$ knockout lines ($n = 10$, combined $Pbrm1$ knockout #4 and #18). The sequencing reads were aligned to mouse reference genome (mm10) with tophat2, and the gene-based read counts were generated by HTSeq. Then the raw counts were normalized with R package DESeq[65]. In this report, we worked with the normalized count data, which includes normalized expression values for 18279 genes.

GSEA was applied to assess whether each set of genes (taken as a single group) has coordinated differences across the two groups (mutation vs wild type)[66]. GSEA was run using the software provided by the Broad Institute at http://software.broadinstitute.org/gsea/index.jsp. The confidence interval plots represent the differences in these genes between the two groups (wild-type vs. $PBRM1$ mutant in TCGA or control knockout vs. $Pbrm1$ knockout in murine tumors). The lines depict 95% confidence intervals, and genes with a significant different between the groups (adjusted $p$-value < 0.05) are marked in orange. The $p$-values were obtained from a two-sample $t$-test on the log2-transformed values, and the resulting $p$-values were adjusted for multiple comparisons using the Benjamini–Hochberg method across the gene set. Inference of T cell and CD8 T-cell immune populations based on gene expression[67].

**Reporting summary.** Further information on research design is available in the Nature Research Reporting Summary linked to this article.

## Data availability

Statistical source data for graphical representations and statistical analysis in Fig. 1(b, d, e, g), 2(a–c), 3(b, c, e, g, h), 4(e, g), 5(a, d–f), 6(a, b, e, f), and supplementary Fig. S1(A, B), S3(B), S4(A–D) and S5(A, B) are provided in PBRM1-immunigenicity-Source data file. Uncropped western blot images are available in Supplementary Figs. 6–8. Patient data from TCGA is available from the TCGA data portal (https://portal.gdc.cancer.gov/). Patient data from ICGC is available through the ICGC data portal (https://dcc.icgc.org/). Data from the IMmotion150 trial were downloaded from European Genome-Phenome Archive (EGA) under accession number EGAS00001002928. Data for pre-malignant murine kidneys were acquired from GEO accession GSE83597. Data for Renca tumors were deposited to The Gene Expression Omnibus (GEO) (GSE145919, https://www.ncbi.nlm.nih.gov/geo/query/acc.cgi?acc=GSE145919). For patients from the MSKCC IMPACT study, survival data for ICB-treated patients was acquired from Samstein et al.[50], and mutation data was downloaded from cBioPortal (https://www.cbioportal.org/)[69]. All other data that support the findings of this study are available from the corresponding author upon reasonable request.

## Code availability

Expression data for TCGA patients either was imported into R using the TCGA2STAT package[64] or imported manually into Matlab (2016b). Mouse sequencing reads were aligned to mouse reference genome (mm10) with tophat2, and the gene-based read counts were generated by HTSeq, then the raw counts were normalized with R package DESeq[65]. RNAseq data from the IMMotion150 trial was quantified using kallisto (v0.44.0)[68]. GSEA was run using the software provided by the Broad Institute at http://software.broadinstitute.org/gsea/index.jsp. Any applicable custom scripts are available from the authors upon request.

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

## Acknowledgements

We acknowledge the TCGA Research network. This work was supported by funding from DOD grant W81XWH-17.1.0307, DOD grant CA160728P1, UT MD Anderson Cancer Center CCSG grant 5 P30 CA016672 (Biostatistics shared resource group) and the Adopt-a-Scientist Foundation. N.S. is a CPRIT Scholar in Cancer Research with funding from the Cancer Prevention and Research Institute of Texas (CPRIT) New Investigator Grant RR160021 and Alfred P. Sloan Research Fellowship FG-2018-10723. Additional funding was provide to D.J.M. by NCI grant K99CA240689.

## Author contributions

E.J. provided funding support, supervised the study, wrote the paper and made final decisions on the paper. X.D.L. generated knockdown and knockout cell lines, conceived and supervised the project, prepared figures and wrote the paper. W.K. designed and performed western blot, real-time PCR, ChIP, and ELISA experiments, and assisted manuscript revision. A.H. and T.L. processed tissues and performed IHC and Opal staining. X.Z. and X.D.L. performed mouse experiments. S.I. and M.M.M. performed single-cell collection and flow cytometry. C.B.P., D.J.M., and H.Z. analyzed datasets. P.G.P., K.E.B., S.M.H., N.S., N.M.T., S.Y.L., and W.K.R. provided material support. All authors discussed the results and edited the paper.

## Competing interests

The authors declare no competing interest.
