## [Peer Review File · Nature Communications]

Reviewers' Comments:

Reviewer #1:

Remarks to the Author:

This article is well organized. It is the first time to illustrate that PBRM1 loss reduces IFN γ -induced expression of chemoattractive signals, T cell infiltration and also IFN γ secretion. Due to the absence of effector T cells, TME is less responsive to ICB.

However, several questions should be noted.

1. The authors disclosed that the reason they investigate IFN γ -STAT1 is the importance of IFN γ signaling in modulating the TME. This is not so persuasive and we recommend that other experimental evidence such as transcriptome sequencing may be more convincing.
2. The authors found that PBRM1 might help BRG1 binding to Ifngr2 promoter. This conclusion is relatively superficial in a scientific program. We strongly recommend the authors to further investigate the reason and the precise binding site to the promoter.
3. The TMA from sunitinib-treated primary RCCs, including 12 samples with wild-type PBRM1 and 10 with PBRM1 mutations. This may cause selection bias among the whole cohort. A consecutive cohort should be recruited in the study to undergo further analysis.

Reviewer #2:

Remarks to the Author:

The manuscript "PBRM1 loss defines a non-immunogenic tumor phenotype associated with immunotherapy resistance in renal cell carcinoma" by Liu et al. reports that PBRM1 loss inhibits the interferon JAK STAT signaling pathway resulting in a less immunogenic tumor microenvironment, which may correlate with lack of response to checkpoint inhibitor therapy in patients with advanced renal cell carcinoma (RCC). This is an interesting manuscript, which sheds light on the potential role of PBRM1 mutations, a common finding in RCC, as related to its effect on kidney cancer therapy. Overall this is an interesting observation with a lot of supportive data, but the manuscript is difficult to follow at times. The following issues should be addressed.

1) In figure 1C, the authors state that decreased STAT1 protein levels result from the decrease in the mRNA levels illustrated in figure 1D. The degree of change is not indicated, and the percentages and how the mRNA and protein correlate should be more fully explored. The kinetics of STAT1 protein and the amount of phospho-protein vs protein do not seem to correlate including the degree of phospho-STAT being more than the protein at the same time (2 hr). Also there is a delay in RNA loss and loss of protein which is not discussed. Although these data are clearly interesting and relevant, they are critical and should be better detailed.

2) In general, it is hard to follow the text in the results section discussion when referring to the figures as there are multiple panels in each of the figures throughout the paper. The Figure legends are often short on detail as well. It would be clearer if the author would indicate which lane or which panel or which figure especially in 1C and 1D that they are referring to.

3) The author states that on page 5 there findings indicate there is a positive feedback loop between STAT1 expression and Jak2- STAT1 activity. This is not entirely clear from the data and difficult to follow.

4) The comparison of Irf1 protein and RNA do not correlate whereas the authors attempt to make this a significant association. This is not exactly clear and needs to be further developed.

5) On page 6 where the authors discuss PBRM loss and mRNA expression of LFNGR1, they report percent declines though in the first figure they did not report any change by percentage. Please be consistent.

6) It is unclear why in figure 2, panel 1A of ifngr1 mRNA expression is not felt to be statistically significantly, whereas the difference is not that dissimilar to other experiments such as the adjacent figure in 1A.

it is considered a statistically significantly difference. The author should show a similar western blot for ifngr1 as they do for ifngr2 if they are trying to prove the point that there is a difference in

the effect of interferon on expression in knockout cells.

7) The CHIP experiment in Figure 2A (panel 1D), which is an important point, should be shown instead of just showing scan results.

8) Figure 3, panel 1A is nearly impossible to read.

9) In figure 4 analysis of human tumors, how many of the tumors had PBRM1 mutations and how many had wild typed? Also, figure 4B is difficult to interpret and difficult to read.

10) Minor comment RENCA is misspelled in figure 1 legend.

In summary, this is an interesting exhaustive study that brings new important data to the field. The authors use an isogenic RENCA model (which may not be best model that represents RCC) demonstrating the effect of PBRM1 mutation on the interferon JAK-STAT pathway leading to a decrease in immune function and potentially resistance to check point inhibitor therapy. The animal studies are somewhat complex and difficult to follow and should be more clearly illustrated so that the reader can more easily follow the results. They do however appear to support the authors' hypothesis but should be more streamlined in presentation. The human data is all retrospective but suggestive of the hypothesis regarding PBRM1 mutations having a decreased response to immunotherapy and perhaps explaining a more robust response to anti-angiogenesis therapy in wild type patients. Nevertheless, these data are only suggestive and retrospective. The animal data is much more clear although some of the changes are moderate even though they are statistically significant. The author's conclusion at the end of the abstract is quite definitive and is an overstatement of the data ("These results indicate that PBRM1 loss reduces IFN γ -STAT1 signaling, generates a less immunogenic TME in murine and human RCC samples, and induces resistance to ICB") such that the reader may think that this now predicts resistance to immunotherapy in patients. Although this is not the case and eloquently discussed by the authors in the discussion section of the manuscript, the reader of the abstract may be misled. Regardless, this is an interesting manuscript, which clearly advances the understating of PBRM1 mutations and will undoubtedly lead to both additional clinical and preclinical studies in kidney cancer.

Reviewer #3:

Remarks to the Author:

In this elegant paper, Authors demonstrated that PBRM1 inactivation leads to impaired Interferon-gamma signaling and to upregulated angiogenesis, thus contributing to the generation of a less immunogenic tumor microenvironment in PBRM1-inactivated tumors. Furthermore, these findings were confirmed across 3 independent patient cohorts, as well as in a murine model.

As a whole, the paper is methodologically sound, adequately conducted from an experimental viewpoints, and well written.

Here are my observations and comments.

MAJOR POINTS

1. Authors demonstrated that PBRM1 loss was associated with upregulated angiogenesis in RCC, while in gastric cancer the situation is turned upside-down, PBRM1 loss being associated with a decreased angiogenesis. Authors thus concluded that "These findings further confirm that PBRM1 influences the TCC TME differently from other tumor lineages". I would suggest the Authors to try to discuss these controversial findings, hypothesizing possible reasons for such a difference.

2. Authors correctly highlighted and discussed the contradictory findings coming from their report, as compared (for example) with another recent reports. Among the hypotheses made, they stated that "... the benefit from prior antiangiogenic therapy could have a continued effect during second-line ICB treatment, or could influence ICB response". This very unlikely in my opinion, for biological and pharmacological reasons. Indeed, it is well known that by day 7 after the withdrawal of potent antiangiogenic agents, implanted Lewis lung carcinomas in mice are fully revascularized, and the pericyte phenotype returned to baseline (Mancuso MR, et al. J Clin Invest 2006;116:2610-21).

3. Regarding the same discussion on existing controversial reports on PBRM1 prognostic and

predictive value in RCC, Authors should speculate on possible differences between anti-PD1 vs anti-PD-L1 antibodies. Indeed, the IMmotion 150 cohort (where atezolizumab has been used) ultimately supported the Authors' findings, while the report from the Dana Farber Group suggesting a positive predictive role of PBRM1 mutations (Brown DA, et al. JAMA Oncol 2019;5:1631-3) has been conducted in patients treated with nivolumab.

4. Authors stated that "... the favorable prognosis of patients with PBRM1 mutations could be interpreted as a predictive effect following ICB treatment". Given also the above, Authors should discuss this point a little bit more, using other examples of such an effect in Oncology.

MINOR POINTS

1. Authors should specify the VHL status of the RENCA cells they used.
2. Within the Introduction, Authors stated that "The treatment landscape for advanced RCC evolved significantly ... with the approval of immune checkpoint -blocking antibodies such as nivolumab, pembrolizumab, atezolizumab and ipilimumab". Here they should acknowledge that not all these agents have been approved for the treatment of mRCC. Indeed, despite having been tested in combination with bevacizumab (or as a single agent) atezolizumab has not been registered for the treatment of mRCC

Reviewers' comments:

Reviewer #1 (Remarks to the Author):

This article is well organized. It is the first time to illustrate that PBRM1 loss reduces IFN γ -induced expression of chemoattractive signals, T cell infiltration and also IFN γ secretion. Due to the absence of effector T cells, TME is less responsive to ICB.

Response: We thank the reviewer for this positive comment.

However, several questions should be noted.

1. The authors disclosed that the reason they investigate IFN γ -STAT1 is the importance of IFN γ signaling in modulating the TME. This is not so persuasive and we recommend that other experimental evidence such as transcriptome sequencing may be more convincing.

Response: We totally agree that the focus on IFN γ signaling was greatly strengthened by our transcriptomic data showing PBRM1 loss in murine tumors (Fig. 3A) and human RCC (Fig. 4A) was associated with reduced expression of genes regulating TME, including genes involved in IFN γ signaling pathway, antigen presentation, T cell recruitment, T cell marker and immunosuppressive factors. Our research trajectory did however follow a thought process that began with our stating the hypothesis that IFN γ related signaling may be dysregulated by PBRM1 loss.

In order to organize our manuscript to flow from cell-based molecular studies to tissue based TME evaluations and ultimately to functional experiments, we described the changes in IFN γ signaling (Fig. 1 and 2) before the transcriptomic sequencing data (Fig. 3 and 4). At the beginning of our Introduction (Page 5, first paragraph), we state "Since IFN γ target genes are involved in T cell infiltration, activation and suppression, and thus modulate the TME, we first compared IFN γ -STAT1 activity in *Pbrm1* proficient and deficient Renca cells." At the same time, we wrote an introductory sentence before our transcriptomic results which states "To expand on our observations, we developed a transcriptomic signature to evaluate tumor immunogenicity in RCC, which is based on profiles predicting clinical response to PD-1 blockade in melanoma and urothelial carcinoma ^{1,2}" (page 6, last paragraph), which we hoped would make our manuscript easier to follow.

2. The authors found that PBRM1 might help BRG1 binding to *Ifngr2* promoter. This conclusion is relatively superficial in a scientific program. We strongly recommend the authors to further investigate the reason and the precise binding site to the promoter.

Response: We have performed additional ChIP experiments to further investigate the PBRM1 dependent function of BRG1 in regulating IFN γ target genes and *Ifngr2*. Ni et al. demonstrated that BRG1 is required for STAT1 binding to IFN γ target promoters, such as *CIITA*, *GBP1*, *IFI27* ³. We hypothesized that PBRM1 loss reduced the expression of IFN γ target genes by impairing the binding of BRG1 and STAT1 to promoters. As we expected, the binding of BRG1 and STAT1 to *Cxcl9* and *Cxcl10* promoters decreased in *Pbrm1* knockout Renca cells (Fig. 2A). Furthermore, BRG1 was reported to be required for the recruitment of transcription factor SP1 for matrix metalloproteinase 2 (*MMP2*) expression ⁴. As we showed before, PBRM1 loss also impaired the binding of BRG1 and transcription factor SP1 to the promoter of *Ifngr2*, a gene that is not induced by IFN γ but important for IFN γ signaling. Our results collectively indicate that PBRM1 regulates the binding of BRG1 and transcription factors to both IFN γ receptor and target genes, which provides a broader mechanistic understanding of the reduced IFN γ target gene expression in PBRM1 deficient cells.

Since here we show a broader PBRM1 and BRG1 dependent regulation profile, we did not further investigate the precise *Ifngr2* promoter binding sites.

3. The TMA from sunitinib-treated primary RCCs, including 12 samples with wild-type *PBRM1* and 10 with *PBRM1* mutations. This may cause selection bias among the whole cohort. A consecutive cohort should be recruited in the study to undergo further analysis.

Response: This is a good suggestion, and assessing a larger cohort is part of our future goals. To confirm our results were not due to selection bias, we also presented the results from another independent cohort (Fig. 4G), which show the same directionality as the sunitinib-treated cohort (Fig. S3B). The proportion of *PBRM1* mutated tumors (10/22=45%) in this unselected cohort is close to that found in large published datasets.

Reviewer #2 (Remarks to the Author):

The manuscript “PBRM1 loss defines a non-immunogenic tumor phenotype associated with immunotherapy resistance in renal cell carcinoma” by Liu et al. reports that PBRM1 loss inhibits the interferon JAK STAT signaling pathway resulting in a less immunogenic tumor microenvironment, which may correlate with lack of response to checkpoint inhibitor therapy in patients with advanced renal cell carcinoma (RCC). This is an interesting manuscript, which sheds light on the potential role of PBRM1 mutations, a common finding in RCC, as related to its effect on kidney cancer therapy. Overall this is an interesting observation with a lot of supportive data, but the manuscript is difficult to follow at times. The following issues should be addressed.

Response: We thank the reviewer for their encouraging feedback. In addition to points addressed below, we specifically re-wrote the paragraph describing Fig. 1 to increase readability.

1) In figure 1C, the authors state that decreased STAT1 protein levels result from the decrease in the mRNA levels illustrated in figure 1D. The degree of change is not indicated, and the percentages and how the mRNA and protein correlate should be more fully explored.

Response: Zha *et al* reported that the expression of *Stat1* itself is controlled by IFN γ via an autocrine mechanism⁵. Since this has been published, we are not aiming to show a correlation between mRNA and protein levels, instead, we used both *STAT1* mRNA expression and protein expression as readouts to show the overall activity of IFN γ -STAT1 signaling, with an emphasis on protein expression as a true readout of functionality. In addition, there is a delay between transcription and translation, and the total mRNA and protein levels are influenced by synthesis and degradation, so we expect the same directionality but not an exact correlation at the percentage level.

The kinetics of STAT1 protein and the amount of phospho-protein vs protein do not seem to correlate including the degree of phospho-STAT being more than the protein at the same time (2 hr).

Response: This is a good observation. This disagreement between total protein and phospho-protein is due to their different individual kinetics. IFN γ -induced STAT1 phosphorylation peaks after a short time duration (2hr), but IFN γ -induced STAT1 expression peaks later (8hr). As for “the degree of phospho-STAT being more than the protein at the same time”, it is because of different exposure time and antibody affinity. For example, STAT1 total protein band is more intense after long exposure time (LE) then short exposure (SE).

Also there is a delay in RNA loss and loss of protein which is not discussed. Although these data are clearly interesting and relevant, they are critical and should be better detailed.

Response: This delay was observed in IRF1, where *Irf1* mRNA increased after a short time period (2hr) and dropped after 8 hr, while such a drop was not obvious at a protein level. We discuss this as follows “Interestingly, the IFN γ -induced *Irf1* mRNA expression was relatively fast and transient, which peaked at 2-hr-treatment and dropped after 8-hr-treatment (Fig. S1 A). However, there was a delay in protein loss since no obvious decrease was observed after 2-hour versus 8-hour treatment (Fig. 1C). We suspect that

IRF1 translation is sustained longer than transcription and/or IRF1 protein is more stable than *Irf1* mRNA.”

2) In general, it is hard to follow the text in the results section discussion when referring to the figures as there are multiple panels in each of the figures throughout the paper. The Figure legends are often short on detail as well. It would be clearer if the author would indicate which lane or which panel or which figure especially in 1C and 1D that they are referring to.

Response: We added more information, especially cell type and IFN γ treatment duration, in figure legends to enhance ability to interpret the figures. There are quite a few panels in each figure, and we have tried to label them as clearly as possible. For western blots (1C, 1F), protein names were labeled on the right of each panel; for real time PCR (1D, 1G), mRNA names were labeled on the left of the Y axis. Considering further numbering or labeling of each lane or panel might make the figures too busy, we did not add more figure annotations and have provided context in the figure legends.

3) The author states that on page 5 there findings indicate there is a positive feedback loop between STAT1 expression and Jak2- STAT1 activity. This is not entirely clear from the data and difficult to follow.

Response: Actually the positive feedback loop between STAT1 expression and Jak2- STAT1 activity has been previously reported⁵, and we just used STAT1 as a IFN γ target to confirm the reduction of IFN γ -STAT1 activity in *PBRM1* knockout cells. To make it easier to follow, we rewrote the whole paragraph.

4) The comparison of *Irf1* protein and RNA do not correlate whereas the authors attempt to make this a significant association. This is not exactly clear and needs to be further developed.

Response: IRF1 (interferon regulatory factor 1) is induced by IFN γ and functions as a cooperating transcription factor with STAT1 for a subgroup of downstream target genes^{6,7}. Here we are not aiming at showing a correlation between mRNA and protein levels, instead, we used both *Irf1* mRNA expression and IRF1 protein expression as readouts to show the activity of IFN γ -STAT1 signaling. In addition, there is a delay between transcription and translation, and the total mRNA and protein levels are influenced by rates of synthesis and degradation, so we expect the same directionality but not exact correlation at a percentage level. We rewrote the paragraph describing these findings to add clarity.

5) On page 6 where the authors discuss *PBRM1* loss and mRNA expression of IFNGR1, they report percent declines though in the first figure they did not report any change by percentage. Please be consistent.

Response: This is a good suggestion, and we changed our description to “*Pbrm1* knockout significantly reduced *Ifngr2* transcription”

6) It is unclear why in figure 2, panel 1A of *ifngr1* mRNA expression is not felt to be statistically significantly, whereas the difference is not that dissimilar to other experiments such as the adjacent figure in 1A. It is considered a statistically significantly difference.

Response: Our statistical analysis of panel 1A shows that *Ifngr1* mRNA expression is not statistically significantly different, and we added statistical results (ns) in figures to make this clear. In figure 1A we showed *Pbrm1* mRNA but not *Ifngr1* mRNA.

The author should show a similar western blot for *ifngr1* as they do for *ifngr2* if they are trying to prove the point that there is a difference in the effect of interferon on expression in knockout cells.

Response: Our point is *Ifngr1* was not influenced by *Pbrm1* knockout although it was slightly decreased by IFN γ treatment. Here we added western blot result for IFNGR1 (Fig. 2D), which further confirmed that *Pbrm1* deficiency reduced IFNGR2 but not IFNGR1 at protein level.

7) The CHIP experiment in Figure 2A (panel 1D), which is an important point, should be shown instead of just showing scan results.

Response: In our study, the immune precipitated DNA was amplified and quantified by real-time PCR instead of regular PCR, and thus we did not need to run a gel and scan. Here the ChIP related data were derived from real-time PCR results. We have now added the quantification method in the manuscript body to avoid confusion and state “We performed ChIP assay to evaluate the protein-DNA interaction, and immunoprecipitated DNA by indicated antibody was amplified and quantified by real-time PCR.”

8) Figure 3, panel 1A is nearly impossible to read.

Response: In order to make it easier to read, we updated Fig. 3A by increasing font size and line size and removed the background grid.

9) In figure 4 analysis of human tumors, how many of the tumors had PBRM1 mutations and how many had wild typed? Also, figure 4B is difficult to interpret and difficult to read.

Response: We indicated tumor numbers (WT and MUT) under each column of the graphs. For figure 4G, we specifically mentioned patient numbers in the manuscript as follows: "To further investigate the immune landscape of *PBRM1* deficient tumors, we stained a tissue microarray (TMA) from 20 untreated RCC patients, including 15 samples with wild-type *PBRM1* and 5 with *PBRM1* mutations, and each sample was triplicated." and now we also added this information to the figure legend. Since each tumor sample is triplicated in the TMA (although a few tissue cores are missing as a result of processing), we indicated the number of intact cores under each column.

In order to make figure 4B easier to read, we updated it by increasing font size.

10) Minor comment RENCA is misspelled in figure 1 legend.

Response: Thank you and we corrected it.

In summary, this is an interesting exhaustive study that brings new important data to the field. The authors use an isogenic RENCA model (which may not be best model that represents RCC) demonstrating the effect of PBRM1 mutation on the interferon JAK-STAT pathway leading to a decrease in immune function and potentially resistance to check point inhibitor therapy. The animal studies are somewhat complex and difficult to follow and should be more clearly illustrated so that the reader can more easily follow the results.

Response: We assessed response to both early (started treatment on day 3 post tumor inoculation) and a delayed anti-PD-1 regimen (started treatment when tumors reached 100-200 mm³), and we agree this can make our experiment look complex. In order to make it easier to follow, we (1) added treatment schemas on Fig. 6A and B, (2) simplified figures by removing *in vivo* individual tumor growth graphs to supplementary figures and just left summarized tumor growth and survival results, and (3) described the related results in a more concise way.

They do however appear to support the authors' hypothesis but should be more streamlined in presentation. The human data is all retrospective but suggestive of the hypothesis regarding PBRM1 mutations having a decreased response to immunotherapy and perhaps explaining a more robust response to anti-angiogenesis therapy in wild type patients. Nevertheless, these data are only suggestive and retrospective.

Response: We agree with this comment and concluded our finding in the abstract as "and a retrospective analysis of IMmotion150 RCC cohort also suggests that *PBRM1* mutation reduces benefit from ICB." and in the results section as "These retrospective analysis results collectively suggest that the inferior outcomes of ICB-treated patients with *PBRM1* mutated tumors was not due to a generally poorer prognosis."

The animal data is much more clear although some of the changes are moderate even though they are statistically significant. The author's conclusion at the end of the abstract is quite definitive and is an overstatement of the data ("These results indicate that PBRM1 loss reduces IFN γ -STAT1 signaling, generates a less immunogenic TME in murine and human RCC samples, and induces resistance to ICB") such that the reader may think that this now predicts resistance to immunotherapy in patients. Although this is not the case and eloquently discussed by the authors in the discussion section of the manuscript, the reader of the abstract may be misled.

Response: In order not to mislead the readers, we changed our abstract to "Our study sheds light on the influence of *PBRM1* mutations on IFN γ -STAT1 signaling and TME, and can inform additional preclinical and clinical studies in RCC. "

Regardless, this is an interesting manuscript, which clearly advances the understating of PBRM1 mutations and will undoubtedly lead to both additional clinical and preclinical studies in kidney cancer.

Response: We appreciate such a positive comment.

Reviewer #3 (Remarks to the Author):

In this elegant paper, Authors demonstrated that PBRM1 inactivation leads to impaired Interferon-gamma signaling and to upregulated angiogenesis, thus contributing to the generation of a less immunogenic tumor microenvironment in PBRM1-inactivated tumors. Furthermore, these findings were confirmed across 3 independent patient cohorts, as well as in a murine model. As a whole, the paper is methodologically sound, adequately conducted from an experimental viewpoints, and well written.

Response: We appreciate such a positive comment.

MAJOR POINTS

1. Authors demonstrated that PBRM1 loss was associated with upregulated angiogenesis in RCC, while in gastric cancer the situation is turned upside-down, PBRM1 loss being associated with a decreased angiogenesis. Authors thus concluded that "These findings further confirm that PBRM1 influences the TCC TME differently from other tumor lineages". I would suggest the Authors to try to discuss these controversial findings, hypothesizing possible reasons for such a difference.

Response: We appreciate this suggestion, and we discussed these controversial findings as "PBRM1 has been shown to have pleiotropic effects at a molecular level, including the recognition of acetylated histones and p53, while different bromodomains can either enhance or attenuate nucleosome interaction^{8,9}. Tissue specific cells are programmed to express a set of genes unique to that cell type, and each tumor type has additional canonical mutations specific to that tumor. These varied complex interactions can impact positive and negative feedback loops in a way that produce paradoxical effects."

2. Authors correctly highlighted and discussed the contradictory findings coming from their report, as compared (for example) with another recent reports. Among the hypotheses made, they stated that "...the benefit from prior antiangiogenic therapy could have a continued effect during second-line ICB treatment, or could influence ICB response". This very unlikely in my opinion, for biological and pharmacological reasons. Indeed, it is well known that by day 7 after the withdrawal of potent antiangiogenic agents, implanted Lewis lung carcinomas in mice are fully revascularized, and the pericyte phenotype returned to baseline (Mancuso MR, et al. J Clin Invest 2006;116:2610-21).

Response: We added this point to our discussion to make it more comprehensive. However, in addition to vascularization, different labs have found that RCC samples treated with antiangiogenic agents demonstrated more infiltrating T cells, which potentially could influence ICB response. We thus state "Although tumors revascularized 7 days after withdraw of antiangiogenic agents¹⁰, but tumors received antiangiogenic therapy were associated with increase T cell infiltration, which is predicted to potentiate response to subsequent ICB treatment^{11,12}"

3. Regarding the same discussion on existing controversial reports on PBRM1 prognostic and predictive value in RCC, Authors should speculate on possible differences between anti-PD1 vs anti-PD-L1 antibodies. Indeed, the IMmotion 150 cohort (where atezolizumab has been used) ultimately supported the Authors' findings, while the report from the Dana Farber Group suggesting a positive predictive role of PBRM1 mutations (Brown DA, et al. JAMA Oncol 2019;5:1631-3) has been conducted in patients treated with nivolumab.

Response: This is a very good point and we added the following statement in the discussion: "Third, PBRM1 mutations were associated with resistance to atezolizumab but with response to nivolumab¹³⁻¹⁵, which are antibodies against PD-L1 and PD-1, respectively. PD-1 blockade blocks its interaction with not only PD-L1 but also PD-L2. Meanwhile, PD-L1 can also promote cancer cell survival via PD-1 independent

pathways¹⁶. So, we cannot rule out the possibility that tumors harboring the same mutation may respond to anti-PD1 and anti-PD-L1 antibodies differently.”

4. Authors stated that "... the favorable prognosis of patients with PBRM1 mutations could be interpreted as a predictive effect following ICB treatment". Given also the above, Authors should discuss this point a little bit more, using other examples of such an effect in Oncology.

Response: Thanks for this comment. In RCC, there is a positive correlation between lower risk level as identified IMDC/MSKCC criteria and response to antiangiogenic therapy, where the prognostic and predictive nature of IMDC/MSKCC are interwoven. We have included a statement describing this in the discussion.

MINOR POINTS

1. Authors should specify the VHL status of the RENCA cells they used.

Response: VHL status was specified in results section as "Renca is a broadly used murine RCC cell line, derived from a spontaneously arising tumor in a BALB/c background, and without known *Vhl* and *Pbrm1* mutations."

2. Within the Introduction, Authors stated that "The treatment landscape for advanced RCC evolved significantly ... with the approval of immune checkpoint -blocking antibodies such as nivolumab, pembrolizumab, atezolizumab and ipilimumab". Here they should acknowledge that not all these agents have been approved for the treatment of mRCC. Indeed, despite having been tested in combination with bevacizumab (or as a single agent) atezolizumab has not been registered for the treatment of mRCC

Response: We appreciate this comment, and to be more accurate, we removed atezolizumab.

1. Ayers, M., *et al.* IFN-gamma-related mRNA profile predicts clinical response to PD-1 blockade. *J Clin Invest* **127**, 2930-2940 (2017).
2. Sharma, P., *et al.* Nivolumab in metastatic urothelial carcinoma after platinum therapy (CheckMate 275): a multicentre, single-arm, phase 2 trial. *Lancet Oncol* **18**, 312-322 (2017).
3. Ni, Z., *et al.* Apical role for BRG1 in cytokine-induced promoter assembly. *Proc Natl Acad Sci U S A* **102**, 14611-14616 (2005).
4. Ma, Z., *et al.* Brg-1 is required for maximal transcription of the human matrix metalloproteinase-2 gene. *J Biol Chem* **279**, 46326-46334 (2004).
5. Zha, Z., *et al.* Interferon-gamma is a master checkpoint regulator of cytokine-induced differentiation. *Proc Natl Acad Sci U S A* **114**, E6867-E6874 (2017).
6. Ramsauer, K., *et al.* Distinct modes of action applied by transcription factors STAT1 and IRF1 to initiate transcription of the IFN-gamma-inducible *gbp2* gene. *Proc Natl Acad Sci U S A* **104**, 2849-2854 (2007).
7. Negishi, H., *et al.* Evidence for licensing of IFN-gamma-induced IFN regulatory factor 1 transcription factor by MyD88 in Toll-like receptor-dependent gene induction program. *Proc Natl Acad Sci U S A* **103**, 15136-15141 (2006).
8. Slaughter, M.J., *et al.* PBRM1 bromodomains variably influence nucleosome interactions and cellular function. *J Biol Chem* **293**, 13592-13603 (2018).
9. Cai, W., *et al.* PBRM1 acts as a p53 lysine-acetylation reader to suppress renal tumor growth. *Nat Commun* **10**, 5800 (2019).
10. Mancuso, M.R., *et al.* Rapid vascular regrowth in tumors after reversal of VEGF inhibition. *J Clin Invest* **116**, 2610-2621 (2006).
11. Liu, X.D., *et al.* Resistance to Antiangiogenic Therapy Is Associated with an Immunosuppressive Tumor Microenvironment in Metastatic Renal Cell Carcinoma. *Cancer Immunol Res* **3**, 1017-1029 (2015).

12. Wallin, J.J., *et al.* Atezolizumab in combination with bevacizumab enhances antigen-specific T-cell migration in metastatic renal cell carcinoma. *Nat Commun* **7**, 12624 (2016).
13. Miao, D., *et al.* Genomic correlates of response to immune checkpoint therapies in clear cell renal cell carcinoma. *Science* **359**, 801-806 (2018).
14. Braun, D.A., *et al.* Clinical Validation of PBRM1 Alterations as a Marker of Immune Checkpoint Inhibitor Response in Renal Cell Carcinoma. *JAMA Oncol* (2019).
15. McDermott, D.F., *et al.* Clinical activity and molecular correlates of response to atezolizumab alone or in combination with bevacizumab versus sunitinib in renal cell carcinoma. *Nat Med* **24**, 749-757 (2018).
16. Escors, D., *et al.* The intracellular signalosome of PD-L1 in cancer cells. *Signal Transduct Target Ther* **3**, 26 (2018).

Reviewers' Comments:

Reviewer #1:

Remarks to the Author:

Interesting observation and intact work by Liu et al. which showed a potential mechanism of treatment refractory to ICB in RCC patients. The results are novel and of clinical importance by leading to possible immunotherapies which may improve the outcome and management of PBRM1 mutated tumors. In general, the paper is well written and methodologically sound. In this revised manuscript, the author addressed the comments appropriately with adequate supportive data. However, a further question should be noted which may improve its integrality.

1. The authors mentioned PBRM1 might help BRG1 binding to Ifngr2 promoter, therefore, it is reasonable to postulate that PBRM1 mutation may function as a part of the SWI/SNF complex to reduce the combination of BRG1 to Ifngr2 promoter. Could inhibition of SWI/SNF complex in PBRM1 wt cell model reduce BRG1 and Ifngr2 promoter combination as well?

Reviewer #2:

Remarks to the Author:

The authors have addressed the reviewers concerns and comments.

Reviewer #3:

Remarks to the Author:

Personally, I am fully satisfied by the way Authors answered to the points raised by all Reviewers, making their manuscript suitable for publication

Reviewer #1 (Remarks to the Author):

Interesting observation and intact work by Liu et al. which showed a potential mechanism of treatment refractory to ICB in RCC patients. The results are novel and of clinical importance by leading to possible immunotherapies which may improve the outcome and management of PBRM1 mutated tumors. In general, the paper is well written and methodologically sound. In this revised manuscript, the author addressed the comments appropriately with adequate supportive data. However, a further question should be noted which may improve its integrality.

Response: We appreciate the positive feedback.

1.The authors mentioned PBRM1 might help BRG1 binding to *Ifngr2* promoter, therefore, it is reasonable to postulate that PBRM1 mutation may function as a part of the SWI/SNF complex to reduce the combination of BRG1 to *Ifngr2* promoter. Could inhibition of SWI/SNF complex in PBRM1 wt cell model reduce BRG1 and *Ifngr2* promoter combination as well?

Response: It makes sense that “PBRM1 mutation may function as a part of the SWI/SNF complex to reduce the combination of BRG1 to *Ifngr2* promoter”. However, the SWI/SNF complex consists of multiple subunits, and each subunit may play both common and specific roles in the complex. The effects of inhibition of SWI/SNF might depends on the inhibited subunit. The inhibition targeting BRG1 or PBRM1 is more likely to reduce the binding, as was reported that BRG1 inhibition reduced its binding to *CCNB1* and *LTBP2* promoters and REST-chromatin interaction^{1,2}. We revised our manuscript to state “BRG1 inhibition has been reported to reduce BRG1 binding to *CCNB1* and *LTBP2* promoters and decrease repressor element 1-silencing transcription factor (REST)-chromatin interaction. It is conceivable that *PBRM1* mutations may functionally alter the SWI/SNF complex and reduce the binding of BRG1 to the *Ifngr2* promoter.”

Reviewer #2 (Remarks to the Author):

The authors have addressed the reviewers concerns and comments.

Response: We appreciate such a positive comment.

Reviewer #3 (Remarks to the Author):

Personally, I am fully satisfied by the way Authors answered to the points raised by all Reviewers, making their manuscript suitable for publication

Response: We appreciate this positive comment.

1. Li, Z., Xia, J., Fang, M. & Xu, Y. Epigenetic regulation of lung cancer cell proliferation and migration by the chromatin remodeling protein BRG1. *Oncogenesis* **8**, 66 (2019).
2. Ooi, L., Belyaev, N.D., Miyake, K., Wood, I.C. & Buckley, N.J. BRG1 chromatin remodeling activity is required for efficient chromatin binding by repressor element 1-silencing transcription factor (REST) and facilitates REST-mediated repression. *J Biol Chem* **281**, 38974-38980 (2006).